# Mixture Degree-Corrected Stochastic Block Model for Multi-Group Community Detection in Multiplex Graphs

**Noureddine Henka**                                          *noureddine.henka@rte-france.com*
*R&D Departement*
*RTE France*
*Paris France*

**Mohamad Assaad**                                          *mohamad.assaad@centralesupelec.fr*
*L2S, CentraleSupélec*
*Gif-sur-Yvette, France*

**Sami Tazi**                                          *sami.tazi@rte-france.com*
*R&D Departement*
*RTE France*
*Paris France*

**Reviewed on OpenReview:** *https://openreview.net/forum?id=p9KSFrTLx0*

## Abstract

Multiplex graphs have emerged as a powerful tool for modeling complex data structures due to their ability to handle multiple relational layers. Clustering within a multiplex graph can involve merging vertices into communities that are consistent across all layers, grouping similar layers into clusters, or creating overlapping clusters among vertices and layers. However, a multiplex graph may exhibit distinct vertex communities based on the specific layers to which a vertex is connected. This scenario, termed multi-group community detection, significantly enhances the accuracy of clustering processes and aids in the interpretation of partitions. To date, the current literature on state-of-the-art community detection has not extensively addressed this modeling approach. In this paper, we introduce a novel methodology referred to as the "Mixture Degree-Corrected Stochastic Block Model." This generative model, an extension of the widely utilized Degree-Corrected Stochastic Block Model (DCSBM), is designed to cluster similar layers by their community structures while simultaneously identifying communities within each layer's group. We provide a rigorous definition of the model and utilize an iterative technique to perform inference computations. Furthermore, we assess the identifiability of our proposed model and demonstrate the consistency of the maximum likelihood function through analytical analysis. The effectiveness of our method is evaluated using both real-word data sets and synthetic graphs.

## 1 Introduction

Recent technological advancements have precipitated an exponential increase in data accumulation, conducting to the era of big data. This new ecosystem presents novel challenges in exploring and analyzing extensive datasets, as highlighted in recent literature Elgendy & Elragal (2014). Often, data comes from multiple sources, presenting multiple perspectives from diverse sources and features. Such datasets require sophisticated models that capture intricate relationships and interdependencies across different data types and sources Devagiri et al. (2021); Niu et al. (2016). To address the increasing complexity, multiplex graphs have become a relevant asset Hammoud & Kramer (2020); Zweig (2016).

A multiplex graph consists of a set of vertices connected across multiple layers Han et al. (2023); Magnani et al. (2021). Each layer in a multiplex graph uniquely represents a set of edges that models specific similarities

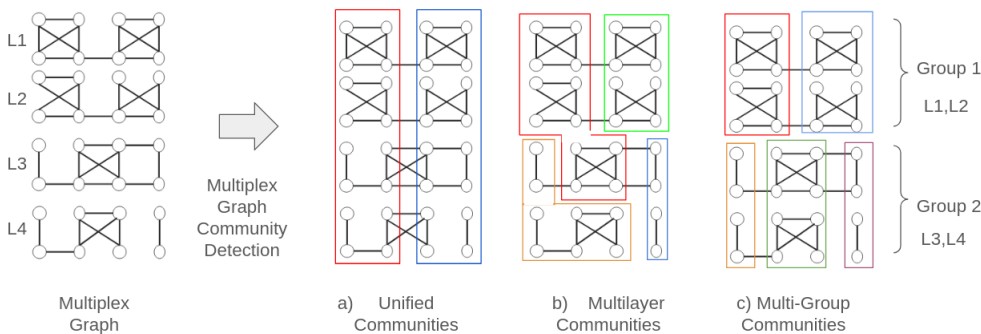

Figure 1: Presentation of a multiplex graph with 4 layers. The results of community detection algorithms on a multiplex graph are categorized into three distinct types of groupings. Each grouping is represented by color-coded boxes, with each color signifying a different community. In figure (a), unified communities are depicted, showing a single community structure shared across all layers. Figure (b) presents a visualization of overlapping communities, where communities extend over several but not all layers. Finally, figure (c) displays multi-group communities, in which similar layers are clustered together, and each cluster maintains a uniform community partition.

between vertices, thereby serving as an effective model for multi-relational data. Additionally, multiplex graphs are adept at modeling time-varying data, offering a robust framework for dynamic data analysis Xia et al. (2020).

Clustering has long been recognized as an effective means for understanding and exploring data across various domains by identifying groups of individuals with strong similarities Fortunato (2010a); Bedi & Sharma (2016). In multiplex graphs, the clustering process poses a significant challenge, yet offers promising solutions for analyzing complex data structures Wang et al. (2019). Community detection in multiplex graphs aims to identify groups characterized by high intra-connectivity and low inter-connectivity Fortunato (2010b). To this end, numerous algorithms employing diverse approaches such as optimization De Meo et al. (2011); Que et al. (2015), spectral computation Li et al. (2018), consensus clustering Mandaglio et al. (2018), and inference Shuo & Chai (2016) have been developed.

The results of these methods can generally be categorized into two groups. The first group optimizes communities to be identical and unified across all layers, which may overlook the inherent diversity within each layer, as presented in (a) from Figure 1. The second approach of multilayer communities allows for overlapping communities across layers and vertices, where vertices may belong to different communities in different layers, as presented in (b) from Figure 1. This model, while computationally intensive, also poses challenges in interpretation and constrains the re-computation of partitions when a new layer arrives, especially as the number of layers increases.

Here, we concentrate on a new outcome of multiplex graph partitioning, named as multi-group clustering. This framework clusters layers into groups, with vertices within each group organized into communities that remain consistent across all layers, as shown in the third clustering outcome in part (c) of Figure 1. This multi-group perspective not only captures the intrinsic structure of each layer but also simplifies the interpretation of the clustering.

Moreover, clustering multiplex layers into groups of similar networks has become an active research area, treating each layer as an individual within a population of networks Mantziou et al. (2023); van der Laan et al. (2022). However, previous approaches have often treated the clustering of layers and the partitioning of vertices independently, neglecting the structural differences in community configurations when layers are clustered. Our dual clustering approach thus aims to enhance both layer clustering and community detection within multiplex graphs, providing a comprehensive solution to the challenges posed by complex network structures.

Although multi-group community detection has not been widely addressed in the literature, its relevance is apparent in real-world datasets. For example, individuals sharing similar musical tastes may not necessarily align in their political or sports affiliations, that proving the structural diversity of layers. In the financial sector, multiplex graphs represent various transaction types within banking systems—such as online, wire, and ATM transactions. Grouping these transactions by similar dynamics and identifying communities within these groups are crucial for detecting anomalous behaviors and preventing fraud. Additionally, as we present in experiment section in the field of neuroscience, the study of brain connectivity through magnetic resonance imaging offers profound insights. Our experiments demonstrate that this approach can effectively group subjects with similar neurological diagnoses, illustrating the practical applications of multi-group community detection in medical research.

We introduce a new approach for Multi-group community detection based on the Stochastic Block Model (SBM), a generative model extensively developed for community detection, which recognizes the presence of communities within a graph by defining the probability of edges between vertices based on the communities to which they belong Lee & Wilkinson (2019). SBM has been applied to various graph types including single-layer Abbe (2017), multiplex Barbillon et al. (2017), and dynamic networks Corneli (2017). The Degree-Corrected Stochastic Block Model (DCSBM), an advancement of SBM, addresses the heterogeneity of vertex degrees within communities, relaxing the assumption of uniform degree distribution Karrer & Newman (2011). The previous work in Stanley et al. (2015) proposed a model for SBM with multiple strata, which bears some resemblance to multi-group community detection. However, their implementation faces significant limitations when applied to real-world scenarios. These limitations arise due to the strong assumption within the SBM model of a uniform distribution within communities, challenges in initialization, and difficulties in determining the optimal number of communities when used in practical applications.

To overcome the limitations of the SBM models mentioned above, we introduce the Mixture Degree-Corrected Stochastic Block Model (MDCSBM), an innovative framework designed for the multi-group partitioning of multiplex graphs. Our method strategically **groups similar layers and allocates the vertices within these layers into distinct blocks**, as illustrated in part (c) of Figure 1. We posit that layers grouped together adhere to a uniform DCSBM distribution. The degree-corrected feature of our model allows us to relax the constraints of uniform distribution within communities, thus extending the scope of application of our model. To implement this dual clustering process, we utilize the Expectation Maximization (EM) algorithm (in particular we consider Classifier EM (CEM) in this case Celeux & Govaert (1992))to determine the layer groupings, and the Variational Expectation Maximization (VEM) technique for assigning vertices to specific blocks within each group. We also present a theoretical proof confirming the identifiability of the model and establish the consistency of the maximum likelihood estimates derived from this model. To achieve rapid and accurate result, we propose a novel and efficient initialization method based on spectral representation. The experimental section of the paper evaluates the model's performance and scalability across multiple synthetic graphs with multiple layers. We further validate our model's efficacy using two real-world datasets, providing interpretations of the results and highlighting the advantages of our approach over existing models. Throughout the paper, we define "partitioning" as the process of forming communities of vertices, and "clustering" as the method of grouping layers.

## 2 Related Works

Community detection in multiplex graphs can yield diverse types of partitions. For instance, some approaches employ spectral representation-based algorithms and tensor-based algorithms that fuse layers into a centroid graph. This graph is constrained to have $K$ connected components, each representing a community, where $K$ is the number of clusters aimed at capturing highly consistent communities Kang et al. (2019); Wang et al. (2019); Papalexakis et al. (2016). Other methods focus on consensus clustering, which involves computing communities for each layer individually and then identifying the most consistent community across all layers for multiplex community detection Berlingerio et al. (2013); Tagarelli et al. (2017); Tang et al. (2012). Additionally, algorithms originally designed for mono-layer community detection have been adapted for multiplex graphs, often allowing for overlapping between vertices and layers to form multilayer communities, such as Generalized Modularity and Multilayer Infomap Mucha et al. (2010); Afsarmanesh & Magnani (2016); De Domenico et al. (2015); Wilson et al. (2017).

Recent research has also focused on identifying clusters of similar layers that may exhibit the same structure van der Laan et al. (2022). In this approach, each graph is treated as an individual within a population of networks. Various algorithms aim to identify a set of similar layers using methods such as mixture parametric models Mantziou et al. (2023); Kemp et al. (2006), nonparametric methods derived from minimum code length descriptions Kirkley et al. (2023); Kirkley & Newman (2022), models based on graph distance measurements La Rosa et al. (2015), and latent space models Young et al. (2022).

However, these models, which aim to partition the graph into communities, often overlook the potential for multiple clusters within the multiplex graph and do not consider the similarity of communities between different layers. This dual consideration is crucial for enhancing the accuracy and comprehensibility of results, especially as the number of layers increases. In the filed of tabular data, the dual clustering of both features and samples—referred to as co-clustering—has been extensively explored and has proven beneficial for in-depth data analysis Ailem et al. (2015); Nadif & Govaert (2010). Thus, bridging this gap in graph-type data is crucially relevant.

The Stochastic Block Model (SBM) is a generative probabilistic model widely used for community detection in networks. Originally developed for single-layer networks, SBM has been extended to accommodate multi-layer clustering. A notable example is the introduction of the Multi-Layer Stochastic Block Model (MLSBM), which incorporates various types of layer aggregation Vallè s-Català et al. (2016). In contrast, a directed graph model with layers generated independently from the same distribution was explored in De Bacco et al. (2017). Furthermore, Paul & Chen (2016) adapted SBM by assigning each layer its own affinity matrix, yet constrained these matrices to yield consistent communities across layers. Additional developments include the work of Amini et al. (2024); Roy et al. (2006), which focuses on a hierarchical generalization of SBM. This approach seeks to uncover underlying community structures by considering the hierarchical nature of real-world networks. Other adaptations of MLSBM utilize techniques such as Variational Expectation Maximization to infer multi-layer graph structures Barbillon et al. (2017); Corneli et al. (2016); Han et al. (2015). However, these models often grapple with the challenge of exponential parameter scaling, which complicates their application to real-world graphs. Moreover, they generally lack mechanisms for constraining the formation of multi-group structures within the multiplex graph, a critical aspect for capturing the complexity of such networks.

## 3 Mixture DCSBM

This work aims to achieve a dual clustering process by grouping similar layers into a group of layers, and vertices within each group are further clustered into a shared block of vertices. The estimation process of MDCSBM involves determining layer-to-group variables to identify the group of each layer. It is assumed that edges within each group follow a unique DCSBM distribution, treating layers within the same group as independently sampled from the same distribution. Therefore, vertex-to-block variables are estimated for each group to identify the block of each vertex within the same group.

### 3.1 Model Definition

Consider a multiplex graph denoted as $\mathcal{G} = \{G^1, ... G^L\}$ comprising $L$ layer, $G^l = \{V, E^l\}$ represents a single layer, where $l$, s.t $l \in [1, L]$ indicates the layer index, $V$ indicates the set of vertices with $|V| = N$, $E^l$ the set of edges within layer $l$. Let $\mathcal{A} = \{A^1, ..., A^L\}$ be the corresponding adjacency matrix of multiplex graph $\mathcal{G}$, where $A^l$ stands for the adjacency matrix of the graph $G^l$. The underlying graph model in this study is an unweighted and undirected multiplex graph, where the edge distributions follow a Bernoulli distribution. The generalization of this model to a directed graph is straightforward. Finally, an edge $A_{ij}^l$ is defined by a dyad $i, j$, representing its extremity.

Let us consider partitioning multiplex's graph layers into $K$ groups and assume that the vertices of group $k$ are divided into $Q^k$ blocks, where $k \in [1, K]$. The probability of having an edge $A_{i,j}^l$ in layer $l$, within group $k$, giving the block assigned to each vertex, under MDCSBM model is expressed as follows:

$$P(A_{i,j}^l | \mathbf{Z}^k, \mathbf{\Pi}^k, \mathbf{\Theta}^k) = \theta_i^k \theta_j^k \pi_{Z_i^k, Z_j^k}^k \tag{1}$$

where $\mathbf{Z}^k = \{Z_1^k, ...Z_N^k\}$ is the set of vertex-to-block assignments in the group $k$ and $Z_i^k \in \{1, ..., Q^k\}$, $\mathbf{\Theta}^k = \{\theta_1^k, ..., \theta_N^k\}$ is the set of degree heterogeneity parameter for vertices in group $k$, s.t $\theta_i^k > 0$ and $\sum_{i, Z_i^k = q} \theta_i^k = 1 \forall i \in V, \forall q \in [1, Q^k]$. The matrix $\mathbf{\Pi}^k$ has $Q^k \times Q^k$ elements $\pi_{q,w}^k, \forall q, w \in \{1, ..., Q^k\}^2$. Each element represents the probability of an edge existing within the group $k$, depending on the block of its dyad $\{i, j\}$.

Let consider a given layer $G^l$ with known vertex-to-block variable assignments for each group $\mathbf{Z} = \{\mathbf{Z}^1, ..., \mathbf{Z}^K\}$. The probability of existing edge $A_{ij}^l$ between dyad $(i, j)$ from the MDCSBM model, conditioned on $\mathbf{Z}$, can be described as a mixture distribution of $K$ independent DCSBMs, expressed as:

$$
\begin{aligned}
P(A_{ij}^l = 1 | \mathbf{Z}; \boldsymbol{\beta}, \mathbf{\Pi}, \mathbf{\Theta}) &= \sum_{k=1}^{K} \beta^k \theta_i^k \theta_j^k \pi_{Z_i^k, Z_j^k}^k \\
s.t \sum_k \beta^k &= 1, \\
\sum_{i, Z_i^k = q} \theta_i^k &= 1, \forall q \in [1, Q^k], \forall k \in [1, K]
\end{aligned}
\tag{2}
$$

where $\boldsymbol{\beta} = \{\beta^1, ..., \beta^K\}$ denotes the set of probabilities of layer $l$ to be generated from group $k$, representing the mixture weights of MDCSBM, $\mathbf{\Pi} = \{\mathbf{\Pi}^1, \mathbf{\Pi}^2, ..., \mathbf{\Pi}^K\}$ is the set of parameters of each group, and $\mathbf{\Theta} = \{\mathbf{\Theta}^1, \mathbf{\Theta}^2, ..., \mathbf{\Theta}^K\}$ is the set of degree heterogeneity for each group. This model incorporates $K$ distributions from which layers can be generated.

To address the challenge of maximizing the log-likelihood function due to the sum in the mixture model, we introduce a set of latent variables $\mathbf{Y}$ that represents the layer-to-group assignment. Specifically, $y_{lk}$, s.t $l \in [1, L]$ and $k \in [1, K]$, takes the value of one when layer $l$ is generated from group $k$ and zero otherwise. The updated formulation for the probability of an existing edge is as follows:

$$
\begin{aligned}
P(A_{ij}^l = 1 | \mathbf{Y}, \mathbf{Z}; \mathbf{\Pi}, \mathbf{\Theta}) &= \prod_{k=1}^{K} (\theta_i^k \theta_j^k \pi_{Z_i^k, Z_j^k}^k)^{y_{lk}} \\
P(y_{lk} = 1; \beta) &= \prod_{k=1}^{K} (\beta^k)^{y_{lk}}
\end{aligned}
\tag{3}
$$

Additionally, for any group $k$, we identify the probability of a vertex $i$ to be assigned to block $q$ as follows:

$$
\begin{aligned}
P(Z_i^k = q; \boldsymbol{\alpha}^k) &= \alpha_q^k \\
s.t \sum_{q=1}^{Q^k} \alpha_q^k &= 1
\end{aligned}
\tag{4}
$$

such that $\boldsymbol{\alpha} = \{\boldsymbol{\alpha}^1, \boldsymbol{\alpha}^2, ..., \boldsymbol{\alpha}^K\}$ and $\boldsymbol{\alpha}^k = \{\alpha_1^k, \alpha_2^k, ..., \alpha_{Q_k}^k\}$.

Let us consider $\mathbf{\Delta} = \{\mathbf{\Delta}^1, \mathbf{\Delta}^2, ..., \mathbf{\Delta}^K\}$, such that $\mathbf{\Delta}^k = \{\mathbf{\Pi}^k, \mathbf{\Theta}^k, \boldsymbol{\alpha}^k\}$ be the aggregation of group parameters, the log-likelihood of the proposed model is written as follows:

$$
\mathcal{L}(\mathcal{A}, \mathbf{Y}, \mathbf{Z}; \boldsymbol{\beta}, \mathbf{\Delta}) = \sum_{l=1}^{L} \sum_{k=1}^{K} y_{lk} \left[ ln\beta^k + \mathcal{L}(A^l, \mathbf{Z}^k; \mathbf{\Delta}^k) \right]
\tag{5}
$$

where $\mathcal{L}(A^l, \mathbf{Z}^k; \mathbf{\Delta}^k)$ is the complete log-likelihood of layer $l$ in group $k$ with parameters $\mathbf{\Delta}^k$, formulated as follow:

$$
\begin{aligned}
\mathcal{L}(A^l, \mathbf{Z}^k; \mathbf{\Delta}^k) &= ln(P(A^l | \mathbf{Z}^k; \mathbf{\Pi}^k, \mathbf{\Theta}^k)) + ln(P(\mathbf{Z}^k; \boldsymbol{\alpha}^k)) \\
&= \sum_{i,j,i \neq j} A_{ij}^l ln(\pi_{Z_i Z_j}^k) + (1 - A_{ij}^l) ln(1 - \pi_{Z_i Z_j}^k) + \sum_{i,j,i \neq j} ln(\theta_i^k \theta_j^k) + \sum_{i=1} ln(\alpha_{Z_i}^k)
\end{aligned}
\tag{6}
$$

In the context of inferring information from a given multiplex graph, the primary objectives involve assigning each layer to a specific group $k$ using variable $y_{lk}$, then assigning each vertex $i$ within group $k$ to a particular block $q$ using variable $Z_i^k$, and optimizing the parameters $\boldsymbol{\beta}$ and $\boldsymbol{\Delta}$.

The verification of the parameter's identifiability and the assessment of the maximum likelihood consistency are performed as follows.

### 3.1.1 Identifiability

The identifiability of the parameters for uni layer Bernoulli SBM has been proved in Celisse et al. (2012). The proof has been extended to a multiplex graph for pillar division Barbillon et al. (2017). We extend this analysis for multiplex DCSBM with multi groups.

**Theorem 3.1.** *Let assume that there is $K$ groups and every group has the same number of blocks $Q^k = Q^{k'} = Q \ \forall k, k' \in \{1, ..., Q\}^2$. Assume for any $q \in \{1, ..., Q\}, k \in \{1, ..., K\}$, $\alpha_q^k > 0, \beta^k > 0$. Let $\boldsymbol{\Pi} \in ]0, 1[^{K*Q \times K*Q}$ diagonal block that contains matrices $\Pi^k$ at diagonal as follow:*

$$\begin{bmatrix} \boldsymbol{\Pi^1} & ... & 0 \\ \vdots & ... & \vdots \\ 0 & ... & \boldsymbol{\Pi^K} \end{bmatrix}$$

*Let also $\boldsymbol{\alpha}$ be a $K*Q \times K*Q$ matrix, which is the diagonilization of $[\alpha_1^1, ...\alpha_Q^1, ...\alpha_Q^K]$ vector, and $\boldsymbol{\beta}$ be a $K*Q \times K*Q$ matrix, which is the diagonilization of $[\beta^1, ...\beta^1, \beta^2...\beta^K]$ vector, where $\beta^i$ is repeated $Q$ times, $\forall i \in \{1, ..., K\}$. Assume that the elements of $\boldsymbol{r} = \boldsymbol{\Pi}.\boldsymbol{\alpha}.\boldsymbol{\beta}$ are distinct. Then the MDCSBM parameters are identifiable under equivalent solutions.*

The proof is presented in the supplementary material in A

### 3.1.2 Consistency of the Maximum Likelihood

Let us assume that the following assumptions hold:

**Assumption 3.2.** *For every $q \neq q'$ ,there exists $w \in \{1, ..., Q^k\}$ such that $\pi_{qw}^k \neq \pi_{q'w}^k$, or $\pi_{wq}^k \neq \pi_{wq'}^k$*

**Assumption 3.3.** *There exists $\zeta > 0$ such that $\forall (q, w) \in \{1, ..., Q^k\}$, $\pi_{qw}^k \in ]0, 1[ \rightarrow \pi_{qw} \in [\zeta, 1 - \zeta]$*

**Assumption 3.4.** *There exists $\gamma \in 1/Q^k$ such that $\forall q \in \{1, ..., Q^k\}$, $\alpha_q^k \in ]0, 1[ \rightarrow \alpha_q \in [\gamma, 1 - \gamma]$*

**Assumption 3.5.** *There exists $\xi \in 1/K$ such that $\forall k \in \{1, ..., K\}$, $\beta^k \in ]0, 1[ \rightarrow \beta^k \in [\xi, 1 - \xi]$*

**Theorem 3.6.** *Let $(\Theta, d)$ and $(\Psi, d')$ denote metric spaces and let $M_n : \Theta \times \Psi \rightarrow \mathcal{R}$ be a random function and $M : \Theta \rightarrow \mathcal{R}$ a deterministic function such that for every $\epsilon > 0$*

$$sup_{d(\theta, \theta_0)} M(\theta) < M(\theta_0) \tag{7}$$

$$sup_{(\theta, \psi) \in \Theta \times \Psi} |M_n(\theta, \psi) - M(\theta)| := ||M_n - M||_{\Theta \times \Psi} \rightarrow 0 \tag{8}$$

*and $(\hat{\theta}, \hat{\psi}) = \underset{\theta, \psi}{argmax} \, M_n(\theta, \psi)$, then*

$$d(\hat{\theta}, \theta_0) \rightarrow 0 \tag{9}$$

The proof is presented in the supplementary material in B.

## 4 Optimization of Log Likelihood Function

As explained previously, the MDCSBM depends on layer-to-group and vertex-to-block assignment variables. We set an iterative approach to address this joint assignment clustering challenge. To elaborate, we estimate layer-to-group assignment variables by utilizing the Expectation Maximization (EM) technique. Then, to infer the DCSBM model within each group, we use the Variational EM (VEM) technique. This technique proves its performance in maximizing DCSBM distribution parameters while estimating the latent vertex-to-block variables.

### 4.1 Estimation of Layer-to-Group Variables

The computation of layer-to-group latent variables estimation is derived from equation 5, which defines the complete log-likelihood. The estimation process involves calculating the expectation of the log-likelihood based on the posterior distribution of layer-to-group latent variables, and it can be expressed as follows:

$$E_{\mathbf{Y}}[\mathcal{L}(\mathcal{A}, \mathbf{Y}, \mathbf{Z}; \boldsymbol{\beta}, \boldsymbol{\Delta})] = \sum_{l=1}^{L} \sum_{k=1}^{K} E(y_{lk}) \Big[ ln\beta^k + \mathcal{L}(A^l, \mathbf{Z^k}; \boldsymbol{\Delta}^k) \Big] \tag{10}$$

where $E(y_{lk})$ is the posterior expectation probability of layer $l$ to be generated from group $k$, defined as $p(y_{lk}|A^l, \mathbf{Z}^k)$. Using Bayes theorem, the estimation of layer-to-group is computed as follows:

$$E(y_{lk}) = \frac{\beta^k P(A^l, \mathbf{Z}^k | \boldsymbol{\Delta}^k)}{\sum_j \beta^j P(A^l, \mathbf{Z}^j | \boldsymbol{\Delta}^j)} \tag{11}$$

where $P(A^l, \mathbf{Z}^k; \boldsymbol{\Delta}^k)$ is written as follows:

$$P(A^l, \mathbf{Z}^k; \boldsymbol{\Delta}^k) = P(A^l | \mathbf{Z}^k; \boldsymbol{\Pi}^k, \boldsymbol{\Theta}^k) P(\mathbf{Z^k}; \boldsymbol{\alpha^k}) \tag{12}$$

The estimation of layer-to-group prioritizes the layer that maximizes the likelihood distribution for a specific group, which is presented as the classification step in CEM algorithm. In order to assign each layer to a single group, the selection of the group is based on the following:

$$y_{lk} = \underset{j}{\operatorname{argmax}}\, y_{lj} \tag{13}$$

### 4.2 Maximization of Likelihood Parameters and Vertex-to-Block Variables

Once the layer-to-group assignment is identified, the MDCSBM parameters can be maximized, and the vertex-to-block variable can be estimated too.

#### 4.2.1 Maximization of $\beta$

Considering equation 5, the optimization of $\boldsymbol{\beta}$ involves expressing the complete log-likelihood as follows:

$$\mathcal{L}(\mathcal{A}, \mathbf{Y}; \boldsymbol{\beta}, \boldsymbol{\Delta}) = \sum_{l=1}^{L^k} ln\beta^k + C(\beta^k)$$
$$s.t \sum_{k=1}^{K} \beta^k = 1 \tag{14}$$

where $L^k = \{l \in [1, L], s.t \;\; y_{lk} = 1\}$, set of layer of group $k$, and $C(\beta^k)$ defined as a constant regarding on $\beta^k$. By employing the Lagrange multiplier approach, the solution that satisfies the Karush-Kuhn-Tucker (KKT) conditions can be expressed as follows:

$$\beta^k = \frac{N^k}{N} \tag{15}$$

where $N_k = |L^k|$ is the number of layers in the groups $k$.

#### 4.2.2 Estimation of vertex-to-block and Maximization of Parameter $\Delta_k$

Our mathematical models assume that the layers within the same group are generated independently from the same DCSBM distribution specific to that group. Let $\mathcal{A}^k$ the multiplex graph that contains the layers of group $k$, based on equation 6, the log-likelihood of group $k$ is expressed as follows:

$$L(\mathcal{A}^k, \mathbf{Z}^k; \mathbf{\Delta}^k) = \sum_{l \in L^k} \sum_{i,j,i \neq j} ln(\theta_i^k \theta_j^k) + \sum_{i=1} ln(\alpha_{Z_i}^k) + \sum_{l \in L^k} \sum_{i,j,i \neq j} A_{ij}^l ln(\pi_{Z_i Z_j}^k) + (1 - A_{ij}^l) ln(1 - \pi_{Z_i Z_j}^k)$$

$$s.t \sum_{q}^{Q^k} \alpha_q^k = 1, \sum_{i, Z_i^k = q} \theta_i^k = 1 \tag{16}$$

In order to optimize the parameters that maximize the previous equation, it is essential to first estimate the latent assignment variables. This task is addressed using the Expectation Maximization (EM) algorithm, which requires computing the posterior probability of the latent variable $\mathbf{Z}^k$ with respect to the observed layers, denoted as $P(\mathbf{Z}^k | \mathcal{A}^k)$. However, for single-layer graphs, it has been demonstrated that calculating this conditional probability is computationally intractable Celisse et al. (2012). Various approaches have been proposed in the literature to tackle this challenge Li et al. (2015); Lee & Wilkinson (2019), but they tend to suffer from the curse of dimensionality, mainly when dealing with large-scale datasets.

The Variational EM technique has been adopted to address this issue as an alternative technique for handling DCSBM estimation challenges. Previous studies have established the VEM technique's convergence for single-layer SBM and DCSBM models and multiplex SBM graphs Celisse et al. (2012); Barbillon et al. (2017). The VEM approach involves approximating the posterior distribution $P(\mathbf{Z}^k | \mathcal{A}^k)$, by another distribution $R_{\mathcal{A}^k}$ over $Z^k$. By leveraging this approximation, the marginal log-likelihood over $\mathbf{Z}^k$ can be expressed as follows:

$$\mathcal{L}(\mathcal{A}^k; \mathbf{\Delta}^k) = \sum_{\mathbf{Z}^k} R_{\mathcal{A}^k}(\mathbf{Z^k}) \mathcal{L}(\mathcal{A}^k, \mathbf{Z}^k; \mathbf{\Delta}^k) - \sum_{\mathbf{Z}^k} R_{\mathcal{A}^k}(\mathbf{Z^k}) ln\Big(R_{\mathcal{A}^k}(\mathbf{Z^k})\Big) +$$

$$\mathbf{KL}\big[R_{\mathcal{A}^k}(\mathbf{Z^k}), P(\mathbf{Z}^k | \mathcal{A}^k; \mathbf{\Delta}^k)\big] \tag{17}$$

where $\mathbf{KL}$ is the Kullback-Leibler divergence. Therefore, instead of maximizing $\mathcal{L}(\mathcal{A}^k; \theta^k)$ for the observed data, the VEM technique optimizes a lower bound of $\mathcal{L}(\mathcal{A}^k; \theta^k)$, denoted as $\mathcal{I}_\theta(R_{\mathcal{A}^k})$. This lower bound is known as the evidence lower bound, and it can be defined as follows:

$$\mathcal{I}_\theta(R_{\mathcal{A}^k}) = \mathcal{L}(\mathcal{A}^k; \boldsymbol{\theta}^k) - \mathbf{KL}\big[R_{\mathcal{A}^k}(\mathbf{Z^k}), P(\mathbf{Z}^k | \mathcal{A}^k; \mathbf{\Delta}^k)\big]$$

$$= \sum_{\mathbf{Z}^k} R_{\mathcal{A}^k}(\mathbf{Z^k}) \mathcal{L}(\mathcal{A}^k, \mathbf{Z}^k; \mathbf{\Delta}^k) - \sum_{\mathbf{Z}^k} R_{\mathcal{A}^k}(\mathbf{Z^k}) log R_{\mathcal{A}^k}(\mathbf{Z^k}) \tag{18}$$

$$\leq \mathcal{L}(\mathcal{A}^k, \mathbf{Z}^k; \mathbf{\Delta}^k)$$

The equality between the evidence lower bound and the log-likelihood holds when $R_{\mathcal{A}^k}(\mathbf{Z^k})$ is equal to the true posterior distribution $P(\mathbf{Z}^k | \mathcal{A}^k; \mathbf{\Delta}^k)$. Maximizing the lower bound $\mathcal{I}_{\boldsymbol{\theta}}(R_{\mathcal{A}^k})$ is equivalent to minimizing the Kullback-Leibler divergence $\mathbf{KL}\big[R_{\mathcal{A}^k}(\mathbf{Z^k}), P(\mathbf{Z}^k | \mathcal{A}^k; \mathbf{\Delta}^k)\big]$. Regarding to integer nature of vertex-to-block variables, to approximate the posterior distribution, we select $R_{\mathcal{A}^k}(\mathbf{Z^k})$ as follows:

$$R_{\mathcal{A}^k}(\mathbf{Z^k}) = \prod_{i=1}^{N} h(\mathbf{Z}_i^k; \tau_i^k) \tag{19}$$

where $h(::; \tau_i^k)$ is a multinomial distribution with parameters $\boldsymbol{\tau} = \{\tau_1^k, ... \tau_{Q^k}^k\}$. The entity $\tau_{iq}^k$ approximates the probability that vertex $i$ belongs to the community $q$ in group $k$. The $\mathcal{I}_\theta(R_{\mathcal{G}^k})$ can be written as follows:

$$\mathcal{I}_\theta(R_{\mathcal{A}^k}) = \frac{1}{2} \sum_{l \in L^k} \sum_{i \neq j} \sum_{qw} \tau_{iq}^k \tau_{jw}^k \Big[A_{ij}^l ln(\pi_{qw}^k) + (1 - A_{ij}^l) ln(1 - \pi_{qw}^k)\Big] + \sum_i \sum_q \tau_{iq}^k ln(\alpha_q^k) +$$

$$\sum_i \sum_q \tau_{iq}^k ln(\theta_i^k) - \sum_i \sum_q \tau_{iq}^k ln(\tau_{iq}^k) \tag{20}$$

---

**Algorithm 1** Inference of Likelihood of MDCSBM

---

**Input**: $\mathcal{G}, K, \mathbf{Q} = [Q^1, ..., Q^K]$
**Output**: $\mathbf{Y}, \mathbf{Z}, \mathbf{\Pi}, \mathbf{\Theta}, \boldsymbol{\beta}, \boldsymbol{\alpha}$

   Initialize $\mathbf{Y}, \mathbf{Z}$ with Algorithm 2
   **while** Iteration < Iteration max $\wedge$ Not Converge  **do**
      Estimate $y_{lk}$ with 11
      Compute $y_{lk}$ with 13
      Compute $\alpha_q^k$ with 22
      Compute $\pi_{qw}^k$ with 23
      Compute $\theta_i^k$ with 21
      Compute $\tau_{qw}^k$ with 24
   **end while**

---

The parameters that maximize $\mathcal{I}_\theta(R_{\mathcal{A}^k})$ are derived directly from the previously presented formula. To ensure that the vector $\boldsymbol{\alpha}^k$ and matrix $\mathbf{\Pi}^k$ satisfy the constraints $\sum_q \alpha_q^k = 1$ and $0 \le \pi_{qw} \le 1, \forall q, w \in \{1, ..., Q^k\}^2$, Lagrange multipliers are employed as follows:

$$\hat{\theta}_i^k = \frac{\sum_{l \in L^k} \sum_j A_{ij}^l}{\sum_{l \in L^k} \sum_{i,j \in q} A_{ij}^l} \tag{21}$$

$$\hat{\alpha}_q^k = \sum_i \frac{\tau_{iq}^k}{N} \tag{22}$$

$$\hat{\pi}_{qw}^k = \frac{\sum_{l \in L^k} \sum_{i \ne j} \tau_{iq}^k \tau_{jw}^k A_{ij}^l}{\sum_{l \in L^k} \sum_{i \ne j} \tau_{iq}^k \tau_{jw}^k} \tag{23}$$

$$\hat{\tau}_{iq}^k \propto \hat{\alpha}_q^k \prod_{l \in L^k} \prod_{i \ne j} \prod_w \left[ \hat{\pi}_{qw}^{k \, A_{ij}^l} + (1 - \hat{\pi}_{qw}^k)^{(1 - A_{ij}^l)} \right]^{\hat{\tau}_{jw}^k} \tag{24}$$

where $\hat{\boldsymbol{\alpha}}^k, \hat{\mathbf{\Theta}}^k, \hat{\mathbf{\Pi}}^k, \hat{\boldsymbol{\tau}}^k$ are the best current parameters for the group $k$. Due to the interdependence between $\hat{\Pi}^k$ and $\hat{\tau}^k$, an effective way to determine the best estimation is to alternate between updating $\hat{\Pi}^k$ and $\hat{\tau}^k$ iteratively until convergence. The optimized parameters define the distribution of DCSBM and vertex-to-block assignments for a group $k$. The same computation is executed for each group independently. The overall method is summarized in the algorithm 1.

## 4.3 Initialization model

The initialization process of MDCSBM involves setting up values to layer-to-group variables $\mathbf{Y}$ and vertex-to-block variables $\mathbf{Z}$. Effective initialization of these assignment variables contributes to faster convergence and a higher chance of recovering accurate ground truth values. In the context of mixture models, the K-means algorithm is commonly employed for initializing assignment variables due to its simplicity and quick response. In this paper, we introduce a novel spectral technique that computes layer-to-group and vertex-to-block variables, such that clustering results are used as an initialization for inferring the MDCSBM model.

Consider $\mathbf{U} = \{\mathcal{U}^1, \mathcal{U}^2, ..., \mathcal{U}^K\}$ set of centroid graphs, with each graph $\mathcal{U}^k$ being the centroid that represents the group $k$ . We aim to find layer-to-group variables by optimizing centroids that best represent each group. Then, each centroid $\mathcal{U}^k$ gets clustered into $Q_k$ community to initialize the vertex-to-block assignment variable. One way to find the communities of centroid $k$ is to ensure that it is composed of $Q^k$ disconnected components, where each component corresponds to a community in the graph. In network theory, a graph with a $Q^k$ component exhibits a multiplicity of $Q^k$ null eigenvalues in its corresponding Laplacian matrix. These null eigenvalues are the smallest eigenvalues of the Laplacian matrix. Thus, minimizing the $Q^k$ smallest eigenvalue of the Laplacian matrix associated with $\mathcal{U}^k$ facilitates the formation of $Q^k$ disconnected components within the centroid. Therefore, the model aiming to optimize these representations can be formulated as follows:

$$\min_{\mathcal{U}^1,...,\mathcal{U}^K,\mathbf{F},\mathbf{Y}} \sum_{l=1}^{L}\sum_{k=1}^{K} y_{lk}||\mathcal{U}^k - \mathcal{A}^l||_F^2 + 2\lambda\sum_{k=1}^{K} Tr(\mathbf{F}^{\mathbf{k}^T}\mathbf{L}_{U^k}\mathbf{F}^{\mathbf{k}})$$

$$s.t \ \ \forall i, u_{ij}^k \geq 0, \mathbf{1}^T\mathbf{u}_i^k = 1, \forall k, \tag{25}$$

$$(\mathbf{F}^{\mathbf{k}})^{\mathbf{T}}\mathbf{F}^{\mathbf{k}} = \mathbf{I}, y_{lk} \in \{0,1\}, \sum_{k=1}^{K} y_{lk} = 1$$

where $||.||_F^2$ denote the Frobenius norm, and $u_{ij}^k$ represents element of the centroid $\mathcal{U}^k$, where $\forall i, j \in V$. The Laplacian representation of centroid $\mathcal{U}^k$ is denoted by $L_{\mathcal{U}^k}$, and $\mathbf{F}^k$ represents an embedding vector. The Laplacian matrix is computed in its unnormalized version as follows:

$$L_{\mathcal{U}^k} = D_{\mathcal{U}^k} - \mathcal{U}^k \tag{26}$$

where $D_{U^k}$ denotes the degree matrix, a diagonal matrix. $\mathbf{F}^k \in \mathbf{R}^{N \times Q^k}$ helps to get the number of connected components in the graph. The embedding vector was included in the cost function to relax the non-linear constraint of constructing a centroid with $Q^k$ disconnected components that effectively represent the communities, leveraging the number of null eigenvalues corresponding to the number of components in the graph. The optimization process for this model is described in the next subsection. Experimentally, this initialization helps the MDCSBM to converge faster than random initialization, up to more than 30 times faster, which generally depends on the data structure.

### 4.4 Optimization of initialization model

The initialization model described in Section 4.3 in Equation 25 involves multiple variables, making it challenging to optimize them simultaneously. Therefore, we adopt an iterative technique where each variable will be optimized while the others are held fixed.

#### 4.4.1 Optimizing Y, while U and F are fixed

The model can be represented as follows:

$$\min_{\mathbf{Y}} \sum_{l=1}^{L}\sum_{k=1}^{K} y_{lk}||\mathcal{U}^k - \mathcal{A}^l||_F^2$$

$$s.t \ \ y_{lk} \in \{0,1\}, \sum_{k=1}^{K} y_{lk} = 1 \tag{27}$$

By relaxing the constraint $y_{lk} \in \{0,1\}$ to $y_{lk} \in [0,1]$, the model becomes linear, facilitating the application of analytical solutions that satisfy the Karush-Kuhn-Tucker (KKT) conditions using the Lagrange technique. The analytical solution is expressed as follows:

$$y_{lk} = \frac{||\mathcal{U}^k - \mathcal{A}^l||_F^2}{\sum_{k'=1}^{K}||\mathcal{U}^{k'} - \mathcal{A}^l||_F^2} \tag{28}$$

The determination of the group to which the layer $l$ will be assigned is carried out as follows:

$$y_{lk} = \underset{k}{\operatorname{argmax}} \, y_{lk} \tag{29}$$

#### 4.4.2 Optimizing $\mathcal{U}$, while $Y$ and F are fixed

Firstly, the optimization of centroids $\mathcal{U}^k$ is performed independently, and according to Wang et al. (2020), the objective function $Tr(\mathbf{F}^{\mathbf{k}^T}\mathbf{L}_{U^k}\mathbf{F}^{\mathbf{k}})$ can be expressed as $\sum i, j||\mathbf{f}_i - \mathbf{f}j||_2^2 ui, j$. Therefore, the optimization

for each centroid can be formulated as follows:

$$\min_{\mathcal{U}^k} \sum_{l \in L^k} \sum_{i,j} ||u_{ij} - A_{ij}||_F^2 - \lambda \sum_{i,j} ||\mathbf{f}_i - \mathbf{f}_j||_2^2 u_{i,j} \tag{30}$$
$$s.t \;\; u_{ij} \geq 1, \mathbf{1}^T.\mathbf{u}_i = 1$$

$L_k$ represent the set of layers for which $y_{lk}$ equals one. We denote $||\mathbf{f}_i - \mathbf{f}j||_2^2$ as $dij$. Due to the independence of optimization for each vertex vector $\mathbf{u}_i$ from the others, the optimization of $\mathcal{U}^k$ can be formulated as follows:

$$\min_{\mathbf{u}_i^k} \sum_{l \in L^k} ||\mathbf{u}_i - \frac{\lambda}{2|L^k|}\mathbf{d}_i||_F^2 \tag{31}$$
$$s.t \;\; \forall i, j \;\; u_{ij} \geq 1, \mathbf{1}^T.\mathbf{u}_i = 1$$

The model mentioned above is quadratic with linear constraints, indicating that it is convex. It can resolved using the augmented Lagrange multipliers. Otherwise, any quadratic solver can efficiently resolve this problem.

### 4.4.3 Optimizing $\mathbf{F}^k$ while U and Y are fixed

The optimization of each $\mathbf{F}^k$ for every group is performed independently of the remaining groups. For a given group, the model can be formulated as follows:

$$\min_{\mathbf{F}^k} Tr(\mathbf{F}^{k^T} L_{\mathcal{U}^k} \mathbf{F}^k) \tag{32}$$
$$s.t \;\; \mathbf{F}^{k^T}.\mathbf{F}^k = \mathbf{I}$$

The optimal $\mathbf{F}^k$ can be obtained by extracting $Q^k$ eigenvectors of the Laplacian matrix $L_{\mathcal{U}^k}$, which are associated with the smallest $Q^k$ eigenvalues. It is important to note that $Q^k$ denotes the number of communities within group $k$.

---

**Algorithm 2** Multi Centroids algorithm initialization

---

**Input** $\mathcal{G}, K, \mathbf{Q} = [Q^1, ..., Q^K]$ **Output**: $\mathbf{U}, \mathbf{Y}$
  Initialize $\mathbf{U} = \{\mathcal{U}^1, \mathcal{U}^2, ..., \mathcal{U}^K\}$
  **while** Iteration < Iteration max || Not Converge **do**
    optimize $y_{lk}$ with 29
    optimize $\mathbf{u}_i$ with 31
    compute $\mathbf{F}^k$ from the eigen vector associated to $Q^k$ smallest eigen value of $L_{\mathcal{U}^k}$
  **end while**

---

## 5 Model Selection

To determine the optimal number of groups $K$ and the number of blocks $Q^k$ in each group $k$, we propose using the Bayesian Information Criterion (BIC). The BIC is formulated as follows:

$$BIC = 2\sum_{k=1}^{K}(Q^k)^2 \log(|L^k|N(N-1)) + (N - Q^k)log(|L^k|N) + (Q^k - 1)\log(|L^k|N) \tag{33}$$
$$+2(K-1)\log(L) - \mathcal{L}(\mathcal{G}, \mathbf{Y}, \mathbf{Z}; \boldsymbol{\beta}, \boldsymbol{\Delta})$$

In this expression, $|L^k|$ represents the number of layers associated with group $k$. The BIC helps determining the number of group and blocks by balancing the fit of the model to data penalized by its complexity. The selected model is the one with the lowest BIC.

## 6    Experiments

To assess the properties of the Mixture Degree-Corrected Stochastic Block Model (MDCSBM), we evaluated its performance across various datasets. This analysis includes real-world data on brain connectivity from cerebral imaging and reality mining data concerning proximity interactions among students within a university. Additionally, we utilized multiple synthetic data partitions to explore the limits of MDCSBM and assess its scalability in handling large datasets.

We conducted comparative analyses of MDCSBM against several established algorithms, including the multiplex DCSBM, Generalized Louvain Mucha et al. (2010), and Graph Fusion Spectral Clustering (GFSC) Kang et al. (2019). These algorithms are designed to optimize partitions within multilayer graphs. They are constrained to return a single partition for all multilayer graph, which will be used as the partition for each layer. To quantitatively evaluate the performance of each algorithm, we employed metrics such as Average Normalized Mutual Information (NMI) and Average Adjusted Mutual Information (AMI). These metrics were calculated by computing the NMI and AMI scores for each layer independently, then averaging these scores across all layers. Since the algorithms used for comparison return unified community structures across all layers, the same set of communities is applied in the assessment for each layer. The results presented here reflect only the mean values of these metrics.

### 6.1    Real Data Set

In our real-world application, we examined two datasets: diffusion Magnetic Resonance Imaging (dMRI) and the Reality Mining dataset. To the best of our knowledge, neither dataset includes expert annotations or ground truth labels, meaning there is no reference standard for comparison. Consequently, this lack of ground truth prevents a fair comparison with other algorithms. Instead, we conducted an in-depth analysis of these datasets using our MDCSBM algorithm, offering interpretations and insights based on the results obtained.

#### 6.1.1    Brain Connectivity from Multiplex Graph Representation

To assess the relevance of the MDCSBM in real applications, we conducted a study focusing on brain connectivity using diffusion Magnetic Resonance Imaging (dMRI)[1]. We utilized the HNU1 dataset, which includes 300 undirected graphs derived from 10 brain-scanning sessions across 30 individuals Zuo et al. (2015). Each graph contains 200 vertices representing different regions of the brain, with edges indicating the observed neural connections between these regions. The dataset treats each graph as an independent observation, aligning with methodologies from prior studies Mantziou et al. (2023); Arroyo et al. (2020).

Our primary goal was to detect groups of subjects exhibiting similar brain connectivity patterns and to identify communities within these regions. Considering the brain's division into two hemispheres, each graph inherently comprises two blocks. However, inter-subject scan variations reflect the unique neural states of the individuals.

The MDCSBM was tasked with recognizing subjects with closely related brain scans, accounting for each hemisphere. The model was initialized to distinguish 30 groups, corresponding to the number of subjects, with two blocks per group reflecting the hemispheres. Using the centroid method for initialization, the model consistently identified two blocks for each snapshot across 100 runs, which supports neuroscientific evidence of significant hemispherical independence. However, it also tended to consolidate subjects into 27 distinct groups from the 300 graphs.

Remarkably, the model effectively grouped each subject's graphs within the identified clusters. Specifically, it clustered subjects 8 and 23, 11 and 14, and 10 and 28 together, as detailed in Table 1. The grouping of subjects 11 and 14 aligns with findings from a semi-supervised study Arroyo et al. (2020). Notably, while that study employed a semi-supervised method to achieve these results, our MDCSBM model operates in a completely unsupervised manner. Despite the numerous hyper-parameters that need to be carefully adjusted, our approach still provides more comprehensive results. Additionally, in the study by Mantziou et al. (2023), the experiment limited clustering to only two groups across all layers, which is not optimal for this dataset

---

[1]https://neurodata.io/mri/

containing 30 different subjects. Even with this simplified task, they were unable to consistently group all layers from the same subject within the same cluster. In contrast, our results demonstrate the robustness of the MDCSBM algorithm, as it successfully grouped all layers of each subject into same cluster in every instance.

| Subject | Image | Cluster | Subject | Image | Cluster | Subject | Image | Cluster | Subject | Image | Cluster | Subject | Image | Cluster | Subject | Image | Cluster |
|---|---|---|---|---|---|---|---|---|---|---|---|---|---|---|---|---|---|
| S1 | 1 | C1 | S2 | 1 | C2 | S3 | 1 | C3 | S4 | 1 | C4 | S5 | 1 | C5 | S6 | 1 | C6 |
|  | 2 | C1 |  | 2 | C2 |  | 2 | C3 |  | 2 | C4 |  | 2 | C5 |  | 2 | C6 |
|  | 3 | C1 |  | 3 | C2 |  | 3 | C3 |  | 3 | C4 |  | 3 | C5 |  | 3 | C6 |
|  | 4 | C1 |  | 4 | C2 |  | 4 | C3 |  | 4 | C4 |  | 4 | C5 |  | 4 | C6 |
|  | 5 | C1 |  | 5 | C2 |  | 5 | C3 |  | 5 | C4 |  | 5 | C5 |  | 5 | C6 |
|  | 6 | C1 |  | 6 | C2 |  | 6 | C3 |  | 6 | C4 |  | 6 | C5 |  | 6 | C6 |
|  | 7 | C1 |  | 7 | C2 |  | 7 | C3 |  | 7 | C4 |  | 7 | C5 |  | 7 | C6 |
|  | 8 | C1 |  | 8 | C2 |  | 8 | C3 |  | 8 | C4 |  | 1 | C5 |  | 8 | C6 |
|  | 9 | C1 |  | 9 | C2 |  | 9 | C3 |  | 9 | C4 |  | 9 | C5 |  | 9 | C6 |
|  | 10 | C1 |  | 10 | C2 |  | 10 | C3 |  | 10 | C4 |  | 10 | C5 |  | 10 | C6 |
| S7 | 1 | C7 | S8 | 1 | C8 | S9 | 1 | C9 | S10 | 1 | C10 | S11 | 1 | C11 | S12 | 1 | C12 |
|  | 2 | C7 |  | 2 | C8 |  | 2 | C9 |  | 2 | C10 |  | 2 | C11 |  | 2 | C12 |
|  | 3 | C7 |  | 3 | C8 |  | 3 | C9 |  | 3 | C10 |  | 3 | C11 |  | 3 | C12 |
|  | 4 | C7 |  | 4 | C8 |  | 4 | C9 |  | 4 | C10 |  | 4 | C11 |  | 4 | C12 |
|  | 5 | C7 |  | 5 | C8 |  | 5 | C9 |  | 5 | C10 |  | 5 | C11 |  | 5 | C12 |
|  | 6 | C7 |  | 6 | C8 |  | 6 | C9 |  | 6 | C10 |  | 6 | C11 |  | 6 | C12 |
|  | 7 | C7 |  | 7 | C8 |  | 7 | C9 |  | 7 | C10 |  | 7 | C11 |  | 7 | C12 |
|  | 8 | C7 |  | 8 | C8 |  | 8 | C9 |  | 8 | C10 |  | 8 | C11 |  | 1 | C12 |
|  | 9 | C7 |  | 9 | C8 |  | 9 | C9 |  | 9 | C10 |  | 9 | C11 |  | 9 | C12 |
|  | 10 | C7 |  | 10 | C8 |  | 10 | C9 |  | 10 | C10 |  | 10 | C11 |  | 10 | C12 |
| S13 | 1 | C13 | S14 | 1 | C11 | S15 | 1 | C14 | S16 | 1 | C15 | S17 | 1 | C16 | S18 | 1 | C17 |
|  | 2 | C13 |  | 2 | C11 |  | 2 | C14 |  | 2 | C15 |  | 2 | C16 |  | 2 | C17 |
|  | 3 | C13 |  | 3 | C11 |  | 3 | C14 |  | 3 | C15 |  | 3 | C16 |  | 3 | C17 |
|  | 4 | C13 |  | 4 | C11 |  | 4 | C14 |  | 4 | C15 |  | 4 | C16 |  | 4 | C17 |
|  | 5 | C13 |  | 5 | C11 |  | 5 | C14 |  | 5 | C15 |  | 5 | C16 |  | 5 | C17 |
|  | 6 | C13 |  | 6 | C11 |  | 6 | C14 |  | 6 | C15 |  | 6 | C16 |  | 6 | C17 |
|  | 7 | C13 |  | 7 | C11 |  | 7 | C14 |  | 7 | C15 |  | 7 | C16 |  | 7 | C17 |
|  | 8 | C13 |  | 8 | C11 |  | 8 | C14 |  | 8 | C15 |  | 8 | C16 |  | 8 | C17 |
|  | 9 | C13 |  | 9 | C11 |  | 9 | C14 |  | 9 | C15 |  | 9 | C16 |  | 9 | C17 |
|  | 10 | C13 |  | 10 | C11 |  | 10 | C14 |  | 10 | C15 |  | 10 | C16 |  | 10 | C17 |
| S19 | 1 | C18 | S20 | 1 | C19 | S21 | 1 | C20 | S22 | 1 | C21 | S23 | 1 | C8 | S24 | 1 | C22 |
|  | 2 | C18 |  | 2 | C19 |  | 2 | C20 |  | 2 | C21 |  | 2 | C8 |  | 2 | C22 |
|  | 3 | C18 |  | 3 | C19 |  | 3 | C20 |  | 3 | C21 |  | 3 | C8 |  | 3 | C22 |
|  | 4 | C18 |  | 4 | C19 |  | 4 | C20 |  | 4 | C21 |  | 4 | C8 |  | 4 | C22 |
|  | 5 | C18 |  | 5 | C19 |  | 5 | C20 |  | 5 | C21 |  | 5 | C8 |  | 5 | C22 |
|  | 6 | C18 |  | 6 | C19 |  | 6 | C20 |  | 6 | C21 |  | 6 | C8 |  | 6 | C22 |
|  | 7 | C18 |  | 7 | C19 |  | 7 | C20 |  | 7 | C21 |  | 7 | C8 |  | 7 | C22 |
|  | 1 | C18 |  | 8 | C19 |  | 8 | C20 |  | 8 | C21 |  | 8 | C8 |  | 8 | C22 |
|  | 9 | C18 |  | 9 | C19 |  | 9 | C20 |  | 9 | C21 |  | 9 | C8 |  | 9 | C22 |
|  | 10 | C18 |  | 10 | C19 |  | 10 | C20 |  | 10 | C21 |  | 10 | C8 |  | 10 | C22 |
| S25 | 1 | C23 | S26 | 1 | C24 | S27 | 1 | C25 | S28 | 1 | C10 | S29 | 1 | C26 | S30 | 1 | C27 |
|  | 2 | C23 |  | 2 | C24 |  | 2 | C25 |  | 2 | C10 |  | 2 | C26 |  | 2 | C27 |
|  | 3 | C23 |  | 3 | C24 |  | 3 | C25 |  | 3 | C10 |  | 3 | C26 |  | 3 | C27 |
|  | 4 | C23 |  | 4 | C24 |  | 4 | C25 |  | 4 | C10 |  | 4 | C26 |  | 4 | C27 |
|  | 5 | C23 |  | 5 | C24 |  | 5 | C25 |  | 5 | C10 |  | 5 | C26 |  | 5 | C27 |
|  | 6 | C23 |  | 6 | C24 |  | 6 | C25 |  | 6 | C10 |  | 6 | C26 |  | 6 | C27 |
|  | 7 | C23 |  | 7 | C24 |  | 7 | C25 |  | 7 | C10 |  | 7 | C26 |  | 7 | C27 |
|  | 8 | C23 |  | 1 | C24 |  | 8 | C25 |  | 8 | C10 |  | 8 | C26 |  | 8 | C27 |
|  | 9 | C23 |  | 9 | C24 |  | 9 | C25 |  | 9 | C10 |  | 9 | C26 |  | 9 | C27 |
|  | 10 | C23 |  | 10 | C24 |  | 10 | C25 |  | 10 | C10 |  | 10 | C26 |  | 10 | C27 |

Table 1: The table shows the cluster obtained for each image of each subject by the MDCSBM algorithm. Each of the 30 existing subjects has ten images, which are presented in 10 layers.

### 6.1.2 Reality Mining Study

In this second study, we analyze a physical proximity graph derived from the reality mining study conducted by Eagle and Pentland Eagle & (Sandy), featuring a group of college students and faculty. This dataset comprises undirected graphs representing interactions among 96 students from the Massachusetts Institute of Technology, collected over a nine-month period. Participants were equipped with mobile phones installed with special software that recorded close proximity encounters using Bluetooth technology. Therefore, each layer of this multiplex graph represents a day, with edges indicating at least one observed proximity encounter between individuals on that day. The dataset thus includes $L = 234$ layers, each with $|V| = 96$ vertices.

Figure 2 illustrates the results from the MDCSBM model applied to this dataset. Sub-figure (d) displays the BIC values relative to the number of groups. We identified five distinct groups, with two communities in groups 0 and 2, and one community in groups 1, 3, and 4. The single community structure in groups $1, 3$, and 4 is attributable to the high sparsity of layers within these groups. In contrast, groups like 0 exhibit dual communities as depicted in sub-figure (c), which shows a layer from group 0.

Sub-figures (a) and (b) reveal discernible patterns within these groups. Notably, group 3 (colored red) encompasses layers from weekends and holidays, particularly the Christmas period in December. The other groups capture monthly variations; for instance, group 0 (colored blue) corresponds to the start of the semester, characterized by denser layers. Group 1 (colored orange) represents mid-year, aligning with academic examinations and displaying sparser community interactions. Groups 2 and 4 (colored green and purple, respectively) reflect the end-of-year activities, showing varied layer densities. These interpretations are consistent with typical university schedules, where students primarily interact with classmates during academic sessions but engage with a broader range of acquaintances outside of class times.

Sub-figures (a) and (b) reveal discernible patterns within these groups, particularly for groups 0 and 2, which each contain two communities. From both sub-figures, it is evident that group 0, represented in blue, consists

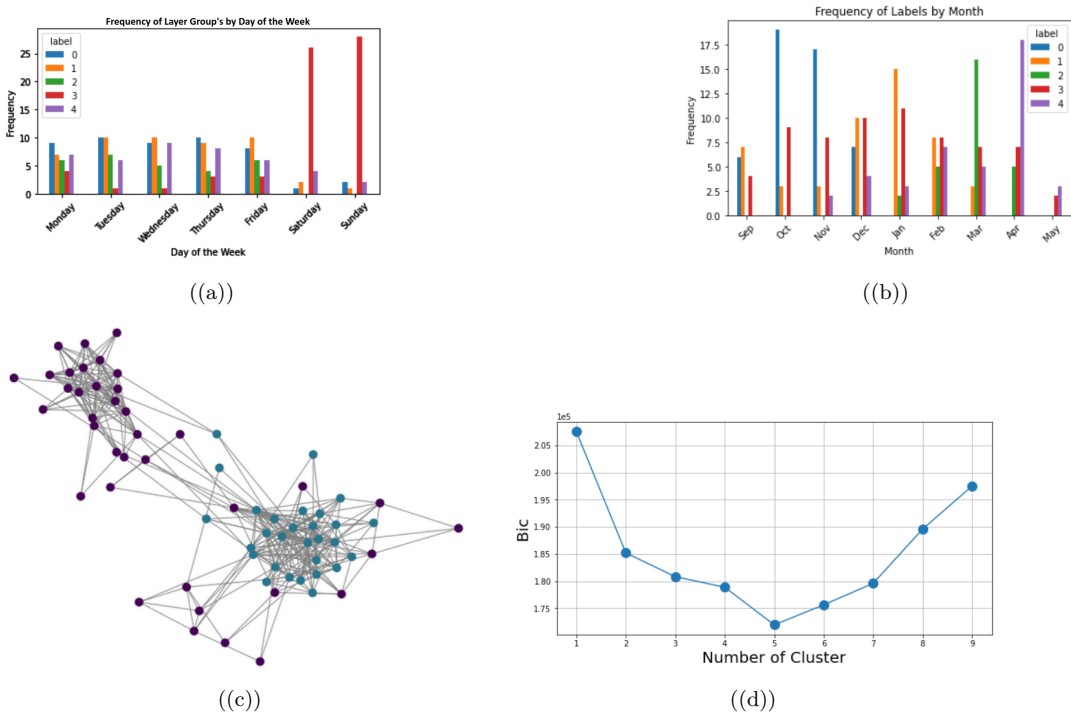

Figure 2: The figure illustrates the outcomes of the Multi-Group clustering applied to the reality mining distance proximity dataset. Panel (a) depicts the total number of days represented in each group. Panel (b) details the monthly distribution of days for each group. Panel (c) presents the community structures within a layer from the group 0. Finally, panel (d) displays the BIC values relative to the number of groups.

of days of the week when courses are likely to occur in the first part of the year. This group is characterized by denser layers, indicating that students are taking two different courses. Conversely, group 2, represented in green, functions similarly to group 0 but pertains to the second part of the year, again involving students taking two different courses. This explains the presence of two communities within these groups.

Group 3, colored red, encompasses layers from weekends and holidays, with a notable focus on the Christmas period in December. Other groups capture monthly variations; for instance, group 1, colored orange, represents the mid-year period, which aligns with academic examinations and displays sparser community interactions. Group 4, colored purple, reflects end-of-year activities, showing varied layer densities. These interpretations align with typical university schedules, where students primarily interact with classmates during academic sessions but engage with a broader range of acquaintances outside of class times.

This experiment demonstrates the effectiveness of the DCMSBM in real-world applications, adeptly handling complex data structures and offering insightful interpretations of dynamic social interactions.

## 6.2 Synthetic Data

### 6.2.1 Variability in Block Size

In this experiment, we aim to assess each group's sensitivity to block size. Therefore, the dataset is composed of 3 groups. Each group of layers consists of 10 graphs, each with 100 vertices organized into four blocks. Vertices inside a block are randomly linked to each other with a probability $\pi_{\text{intra}} = 0.5$, and vertices from different blocks are randomly connected with a probability $\pi_{\text{inter}} = 0.3$. What distinguishes the groups of layers is the number of vertex in each block, $G^1 = \{25, 25, 25, 25\}$, $G^2 = \{20, 25, 25, 30\}$ and $G^3 = \{30, 30, 20, 20\}$, where $G^i$ is the $i^{th}$ group, and for each group, the first number indicates the number of vertex of the first block, the second number the one of the second block, and so forth, as shown in figure 3.

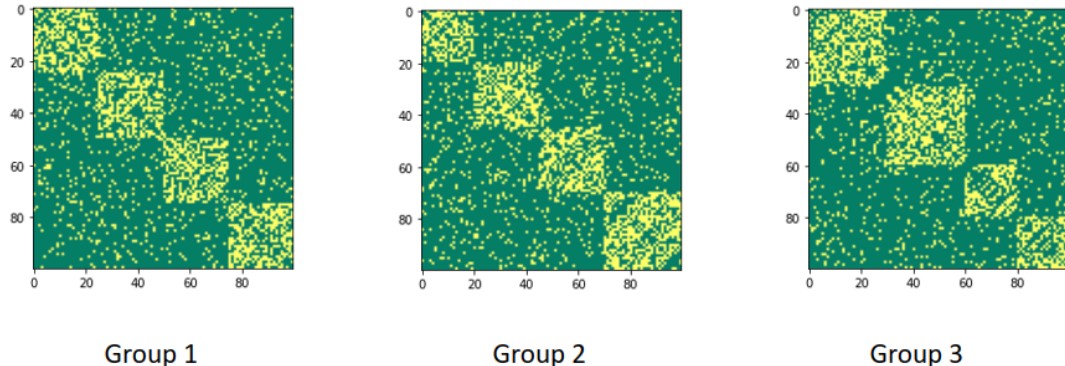

Group 1             Group 2             Group 3

Figure 3: The adjacency matrices correspond to a single layer for each group. $G^1$ is presented on the right, $G^2$ in the middle, and $G^3$ on the left. The intra-block density is set at $\pi_{\text{intra}} = 0.5$. For better visualization, the inter-block connectivity probability is set at $\pi_{\text{inter}} = 0.1$, instead of $\pi_{\text{inter}} = 0.3$ as an experiment.

The actor-based clustering algorithm on multiplex graphs yields an average clustering of vertices. In contrast to MDCSBM, there is no differentiation between blocks of vertices for each group of layers, contributing to the superior NMI/AMI results of MDCSBM over other multiplex algorithms, as shown in Table 2. The mean errors between the predicted and generated parameters of MDCSBM are $mean_{ErroParamMDCSBM} = 0.04$, affirming the improved parameter recovery of MDCSBM compared to the multiplex DCSBM, which results in an error of $mean_{ErroParamMultiDCSBM} = 0.67$.

|      | MDCSBM | DCSBM | GLouvain | GFSC  |
|------|--------|-------|----------|-------|
| NMI  | **100**| 61.40 | 77.44    | 73.80 |
| AMI  | **100**| 58.74 | 75.74    | 72.65 |

Table 2: The NMI and AMI performances on MDCSBM, multiplex DCSBM, Glouvain and GFSC in variability on block size synthetic datasets.

### 6.2.2 Variability in Block Distribution

In this experiment, we aim to assess the model's performance regarding variation in block distribution. Therefore, the dataset for this experiment comprises three groups, each containing ten layers. Each graph consists of 100 vertices distributed across four equally sized blocks ($\{25, 25, 25, 25\}$) with the same intra-community probability ($\pi_{intra} = 0.5$). The distinction between layer groups lies in the probability of having an edge between blocks, denoted as $\pi_{inter}^{G^1} = 0.1$, $\pi_{inter}^{G^2} = 0.3$, and $\pi_{inter}^{G^3} = 0.5$. This variability tests the algorithm's capability to identify groups with different $\boldsymbol{\Pi}$ distributions. Notably, the third group has $\pi_{inter} = \pi_{intra}$, resembling a random graph without communities.

MDCSBM accurately identifies vertex blocks for each group with an estimated parameter error of $mean_{ErroParamMDCSBM} = 0.04$. Additionally, the algorithm recognizes "1 community" for the layers without communities, consistent with a random graph. This result demonstrates the MDCSBM's ability to identify the noisy layer. In contrast, multiplex algorithms exhibit an average effect in the vertex-to-block assignment, especially multiplex DCSBM estimates the vertex-to-block with a higher estimation error ($mean_{ErroParamMultiDCSBM} = 0.71$), as observed previously.

|      | MDCSBM | DCSBM | GLouvain | GFSC  |
|------|--------|-------|----------|-------|
| NMI  | **100**| 55.54 | 66.66    | 53.55 |
| AMI  | **100**| 55.03 | 66.66    | 53.02 |

Table 3: The NMI and AMI performance for MDCSBM, multiplex DCSBM, Glouvain and GFSC in variability on block size synthetic datasets.

### 6.2.3 Variability in Number of Layer

In this experiment, we test the scalability of the method regarding the number of layers. Therefore, we keep the number of vertices, groups, and the block distribution constant while we vary the number of layers for each group. Similar to Experiment 6.2.1, we define three groups with an equal number of layers but distinct block divisions for each group. Specifically, we set $\pi_{intra} = 0.5$ and $\pi_{inter} = 0.3$. The block distribution within each group consists of four blocks, as follows: $G^1 = \{25, 25, 25, 25\}$, $G^2 = \{20, 25, 25, 30\}$, and $G^3 = \{30, 30, 20, 20\}$, where $G^i$ represents the $i^{th}$ group.

The result of the following experiments is shown in the figure 4. We can see that the performance of the MDCSBM in retrieving the optimal blocks for each layer augments when the number of layers augments, compared to the other methods. It can be explained by the law of large numbers, which delineates the convergence in probability to the expected value as the number of samples—the layers in our case—increases. As the layers in the multiplex graph augment, the computation time increases linearly due to the number of parameters scaling linearly with the number of layers, given a fixed number of groups and blocks. This contrasts with other tested algorithms, which do not scale well with large datasets.

### 6.2.4 Variability in Number of Vertices

In this experiment, we assess the scalability of the method regarding the size of each graph. Therefore, we fix the number of layers, the number of groups, and the block distribution of each group, and then we vary the number of vertices of the multiplex graph. We set three groups with an equilibrium of 10 layers for each and different block division such that $G^1 = \{25\%N, 25\%N, 25\%N, 25\%N\}$, $G^2 = \{20\%N, 25\%N, 25\%N, 30\%N\}$,

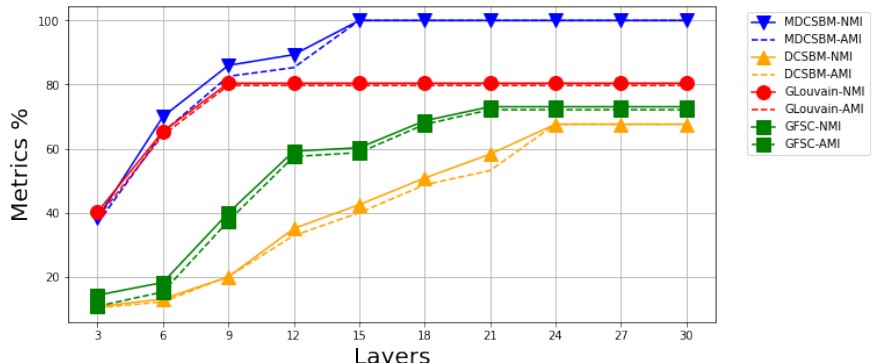

Figure 4: The performance of MDCSBM, DCSBM, Glouvain, and GFSC to find the clusters of vertices regarding the number of layers for each group. The NMI and AMI were used as metrics of performance.

and $G^3 = \{30\%N, 30\%N, 20\%N, 20\%N\}$. The $x\%N$ means $x$ percent from a number of vertices of the layer. The intra-block distribution is $\pi_{intra} = 0.5$ and the inter-distribution is $\pi_{inter} = 0.3$.

In this experiment, we maintain a constant number of layers, groups, and block distribution for each group while varying the graph size by raising its number of vertices. Similar to the previous experiments in 6.2.1, we define three groups with an equal number of layers and distinct block divisions for each.

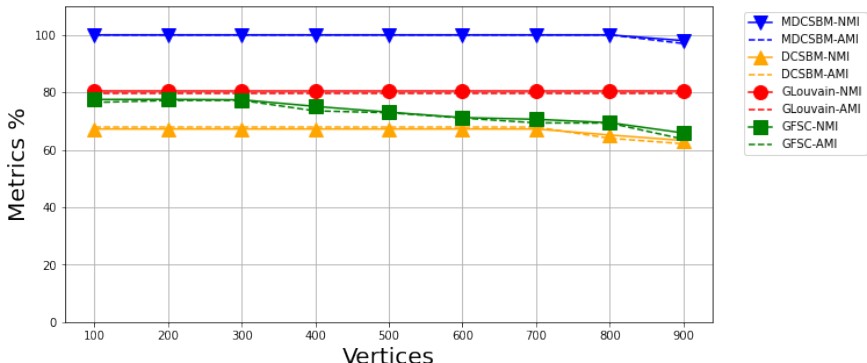

Figure 5: The performance of MDCSBM, DCSBM, Glouvain, and GFSC to find the clusters of vertices regarding the number of vertices in a multiplex graph. The NMI and AMI were used as metrics of performance.

The obtained result is represented in the figure 5. We can notice that the MDCSBM scales to a large dataset with thousands of nodes. The time complexity varies regarding the graph's size and the block's structure. We study the complexity time of the MDCSBM in the supplementary materials.

### 6.2.5 Variability in the Number of Groups

The objective of this experiment is to assess the method's sensitivity with respect to the number of groups within the multiplex graph. Consequently, we keep the number of layers in the multiplex graph fixed at 20, then we vary the number of groups from 2 to 8, ensuring that each group has a distribution distinct from the others.

Figure 6 shows the performance of the algorithms in this experiment. The MDCSBM performs better than the others in finding multigroup community detection. However, we can see that for a high number of groups, the performance of MDCSBM decreases; such a result may be explainable by the reduced number of layers for the high number of groups. We can see that this result will be enhanced when each group's layer number increases.

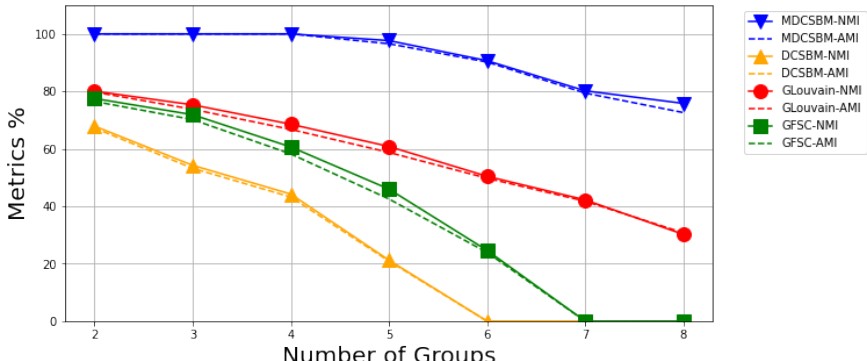

Figure 6: Performance of MDCSBM, DCSBM, GLouvain and GFSC when the number of groups varies in fixed number of layer.

### 6.2.6 Sensitivity to Block Size

In this experiment, we aim to assess the ability of the method to perform in an unbalanced block size. Therefore, we construct a multiplex graph with 20 layers and 2 groups, such that each group has 10 layers of 100 vertices for each, and each group has two blocks with the same distribution. We fix the size of blocks of one group to be 50 vertex per block, and for the other group, we change the block size from 10 to 50 vertex.

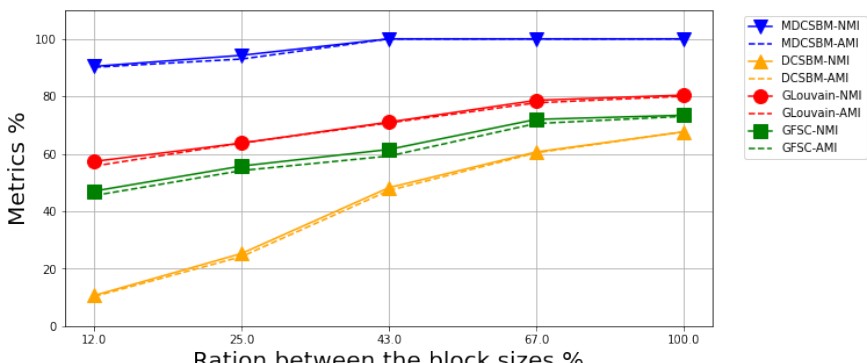

Figure 7: Performance of MDCSBM, DCSBM, GLouvain and GFSC when the size of block variate. The variate group has two blocks; the sensitivity is computed by the fraction between the size of the two blocks.

The performance of the algorithms in this experiment is presented in Figure 6. It is observed that MDCSBM outperforms the others in multi-group community detection. However, with a high number of groups, the performance of MDCSBM decreases, potentially due to the reduced number of layers for the high number of groups. This result is expected to improve when the number of layers in each group increases.

### 6.2.7 Sensitivity to Group Size

This experiment aims to assess the model's performance when the groups have an unbalanced distribution of layers. We consider a multiplex graph with two groups featuring distinct block distributions to achieve this. The number of layers in the first group is fixed at 10, while the number of layers in the second group varies from 1 to 100. Each layer consists of 100 vertices.

Figure 8 illustrates various algorithms' performance in this application. Compared to the others, the MDCSBM algorithm demonstrates notable stable performance, even in scenarios with high imbalance in the number of layers. This result is attributed to the effectiveness of the joint clustering approach, which we believe contributes to achieving accurate results.

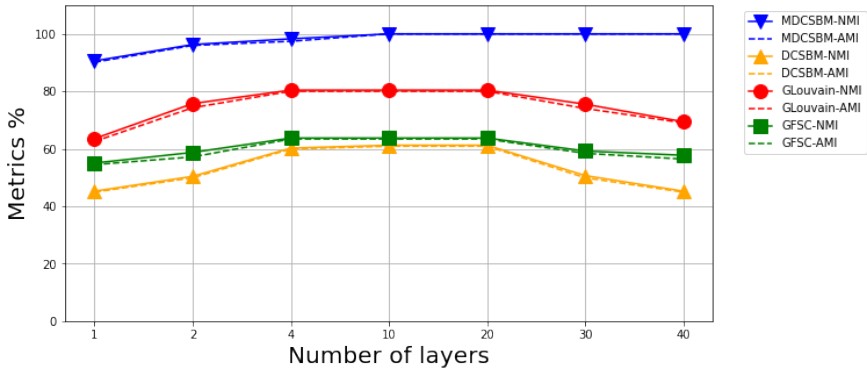

Figure 8: Performance of MDCSBM, DCSBM, GLouvain, and GFSC for different numbers of layers in the group. It varies from 1 to 40 layers.

### 6.2.8 Time Complexity

To illustrate the model's time complexity and scalability to large datasets, we present the time complexity of the model. To achieve this, we conduct two experiments where the time complexity is evaluated for varying numbers of layers and vertices.

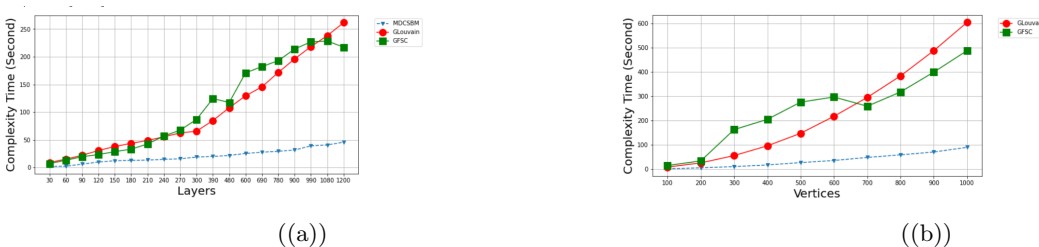

Figure 9: Complexity time regarding a number of layers and vertices. At the right, the time is counted when the number of layers augments. At the left, the time is counted when the number of vertices augments. The time is counted in seconds.

Figure 9 illustrates the time complexity for each case and for each algorithm. Firstly, MDCSBM models scale better in all cases than all other algorithms. Evidently, the time complexity scales linearly with the number of layers. This behavior is attributed to the increase in variables that linearly correlate with the number of layers. Additionally, the time complexity rises more rapidly with an increase in the number of vertices. This outcome results from the significant growth of the combinatorial solution as the number of vertices increases. Despite this, the modeling approach maintains a reasonable time complexity in scenarios with high combinatorial solutions, owing to our efficient initialization process, which facilitates faster convergence compared to random initialization. Furthermore, our initialization accelerates convergence to a favorable local minimum, achieving speeds over 50 times faster than random initialization.

### 6.2.9 Comparison Between MDCSBM and the Initialization Model

The proposed initialization model, based on spectral representation, can be regarded as a comprehensive approach capable of performing multi-group community detection. To evaluate its performance against the MDCSBM model, we used the test set detailed in 6.2.1. This dataset consists of 3 groups, each containing 10 layers, with each layer comprising 100 vertices organized into four blocks. Vertices within the same block are connected randomly with a probability of $\pi_{\text{intra}} = 0.5$, while connections between vertices in different blocks occur with a probability of $\pi_{\text{inter}} = 0.3$. The distinguishing characteristic of each group lies in the distribution of vertices within the blocks: $G^1 = 25, 25, 25, 25$, $G^2 = 20, 25, 25, 30$, and $G^3 = 30, 30, 20, 20$,

where $G^i$ denotes the $i^{th}$ group. Within each group, these values represent the number of vertices in the first, second, third, and fourth blocks, respectively, as depicted in Figure 3.

For this evaluation, we used the MDCSBM algorithm initialized with the proposed spectral model. The experiment was repeated 100 times, and the performance results are summarized in Table 4.

|  | Model Initialization (Spetrale Representation) | MDCSBM |
|---|---|---|
| NMI | 75.48+-22.80 | **95.60 +- 17.33** |
| AMI | 74.60+-23.82 | **95.36+-17.98** |

Table 4: The NMI and AMI performance for MDCSBM and model initialization on block size synthetic datasets.

The superior performance of the MDCSBM model compared to the spectral model can be attributed to the way the MDCSBM utilizes the results produced by the spectral model. The spectral model provides a partition that corresponds to a fixed point of its own cost function, serving as an effective initialization. Although this partition does not necessarily represent a fixed point for the MDCSBM, it enables the MDCSBM to further refine and enhance the clustering results, ultimately achieving better overall performance.

### 6.2.10 Comparison Between Random and Proposed Initialization

The MDCSBM model is highly non-convex, and the resulting outputs are significantly influenced by the initialization of layer-to-group and vertex-to-block values. To address this, we proposed an efficient initialization method based on spectral representation. To illustrate the advantages of this technique, we present a comparison of performance and computation time between random initialization and our proposed spectral-based initialization model.

The same test set described in 6.2.1 was used for this comparison. This dataset comprises 3 groups, each containing 10 layers, with each layer composed of 100 vertices organized into four blocks. Vertices within the same block are randomly connected with a probability of $\pi_{\text{intra}} = 0.5$, while vertices between different blocks are connected with a probability of $\pi_{\text{inter}} = 0.3$. The distinguishing factor among the groups is the number of vertices in each block: $G^1 = 25, 25, 25, 25$, $G^2 = 20, 25, 25, 30$, and $G^3 = 30, 30, 20, 20$, where $G^i$ represents the $i^{th}$ group. Within each group, the numbers correspond to the vertex count of the first, second, third, and fourth blocks, respectively, as illustrated in Figure 3.

We repeated the experiment 100 times and present the performance results and computation times in Table 5.

|  | Model Initialization | Random Initialization |
|---|---|---|
| NMI | **95.60 +- 17.33** | 23.73+-26.40 |
| AMI | **95.36+-17.98** | 22.70+-27.38 |
| Time Complexity (Second) | **3.65+-11.16** | 130.84+-74.79 |

Table 5: The NMI and AMI performance, and time computation for MDCSBM by using the proposed model initialization and random initialization on block size synthetic datasets.

We observed that both the computation time and performance were notably improved when using our proposed initialization method compared to random initialization. This improvement can be attributed to the non-convex nature of the MDCSBM model and its sensitivity to initialization. A robust and well-structured initialization method, such as the one we proposed, is essential for achieving better performance and faster convergence.

### 6.2.11 Model Selection

The defined MSBM model needs a prior knowledge of number of cluster. Ones can think about formulation that allows to optimize the number of cluster in the inference as presented in Roy et al. (2006); Amini et al. (2024). Such models has been based on Chinese Restaurant Process (CRP) to determine the number of

cluster. The CRP modeling in out of the scope of this paper, further enhancement will be reserved for future work. Here in this experiment, we show the results from using he BIC criteria to determine the number of cluster as defined in equation 33. It worth noting to remember that the BIC criteria determine the optimal number of blocks and groups by balancing the fit of the model to the data with a penalization of the model complexity.

In this experiments, we create a multiplex graph of 30 layers. Within this multiplex graph, we set 3 groups of 10 layers for each. We set $\pi_{intra} = 0.5$ and $\pi_{inter} = 0.3$ for all layers. The block distribution within each group consists of four blocks, as follows: $G^1 = \{25, 25, 25, 25\}$, $G^2 = \{20, 25, 25, 30\}$, and $G^3 = \{30, 30, 20, 20\}$, where $G^i$ represents the $i^{th}$ group.

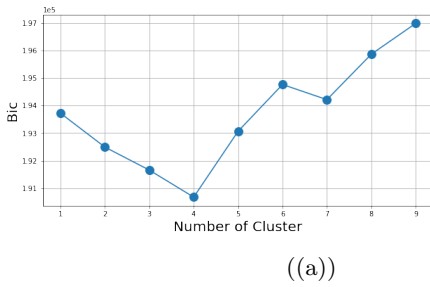 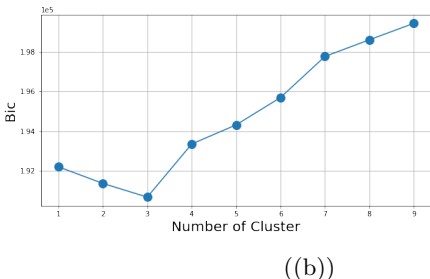

((a))  ((b))

Figure 10: The BIC metric for finding the optimal number of groups and block. At the left, the number of group is fixed to 3, then the BIC is computed for variate number of blocks. At the right, the number of block is fixed to 4, then the BIC is computed for variate number of group.

Figure 10 shows the variation of the Bic regarding the number of blocks when the number of groups is fixed to 3 at the left, and the variation of BIC regarding number of layers when the number of block is fixed to 4 at the right. We can clearly notice that the minimal value of BIC for both experiences indicates the optimal value of cluster as what is attending, showcasing its performance in toy example to define the number of clustering.

## 7 Conclusion

Throughout this paper, we have introduced the Mixture Degree-Corrected Stochastic Block Model (MDCSBM) for multi-group community detection in multiplex graphs. The MDCSBM defines the existing groups that share a similar community structure. Therefore, for each identified group, a distinct DCSBM is derived to ascertain the community structure of each vertex. We have devised an Expectation-Maximization (EM) framework for estimating layer-to-group assignment variables, followed by a Variational EM technique for estimating vertex-to-block assignments. A novel centroid methodology has been proposed to initialize layer-to-group and vertex-to-block variables, enhancing the model's convergence.

This model has been formulated to refine the estimation of the generating model underlying multiplex graphs. It significantly contributes to a better comprehension of community structures within multiplex graphs characterized by multigroups of community memberships. While the current presentation exclusively addresses unweighted graphs, potential extensions encompassing the incorporation of weights through alternative probability distributions such as Gaussian or Poisson distributions exist. Such extensions would undoubtedly enrich the model's applicability in capturing the intricacies of diverse real-world scenarios.

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

# A    Identifiability

*Proof.* We consider the degree node heterogeneity parameter constant and extend the proof from Celisse et al. (2012) to the MDCSBM model as follows. For any group $k$, $r_{q,k}$ is the probability for a giving member from block $q$ in group $k$ to have a connection with another in the same group $r_{q,k} = \sum_{l=1}^{Q} \beta_k \pi_{ql}^k \alpha_l^k$. Let $\mathbf{R}$ be $Q * K$ square matrix such that $R_{i,q,k} = (r_{q,k})^i$ for $i \in 0, ..., Q * k - 1$. $R$ is a Vandermonde matrix that is invertible by assumptions.

Therefore, for any $i = 0, ..., (2Q - 1) * K$, let set

$$\mu_i = \sum_{q,k} \alpha_{q,k}(r_{q,k})^i \tag{34}$$

and $M$ is a $k(Q + 1) \times KQ$ matrix such that

$$M_{ij} = \mu_{i+j} \tag{35}$$

For any $i = 0, ..., Q * k$, we define the Q*K square matrix $M^i$ by removing line $i$ from the matrix. In hence,

$$M^Q = R\boldsymbol{\alpha}R^T \tag{36}$$

where $\boldsymbol{\alpha}$ is $Q * K$ matrix as defined previously, where all $\alpha_q^k \neq 0$. Because $R$ being invertible, then $det(M) > 0$. Let us define

$$B(X, \theta) = \sum_{i=0}^{Q \times K} (-1)^{i+Q*K} det(M^i(\theta))X^i \tag{37}$$

$B$ is of degree $Q \times K$. For $V^i(\theta) = (1, r_i(\theta), ..., (r_i(\theta))^Q)$, then

$$B(r_i(\theta), \theta) = det(M(\theta), V_i(\theta)) \tag{38}$$

The column of $M$ are linearly combinations of $V_i$, then $B(r_i(\theta), \theta) = 0$ for any $i$. It means that $B$ can be factorized as follow:

$$B(x, \theta) = det(M^{Q \times K}) \prod_{i=0}^{KQ-1} (x - r_i(\theta)) \tag{39}$$

Let assume the $\boldsymbol{\theta} = (\mathbf{\Pi}, \boldsymbol{\alpha}, \boldsymbol{\beta})$ and $\boldsymbol{\theta}' = (\mathbf{\Pi}', \boldsymbol{\alpha}', \boldsymbol{\beta}')$ are two sets of parameters such that for any multiplex $\mathcal{G}$ graph with multi-group model, $\mathcal{L}(\mathcal{G}; \boldsymbol{\theta}) = \mathcal{L}(\mathcal{G}; \boldsymbol{\theta}')$. Therefore, we get, $\mu_i(\boldsymbol{\theta}) = \mu_i(\boldsymbol{\theta}')$, that means that

$M^i(\theta) = M^i(\theta')$ for any $i$. The $B(;\theta) = B(;\theta')$ because it dependents on the determinant of $M$, which leads to $r_i(\theta) = r_i(\theta')$. Ths $R(\theta) = R(\theta')$, and

$$\boldsymbol{\alpha}(\theta) = (R(\theta)^T)^{-1}M^{Q,K}R(\theta) = \boldsymbol{\alpha}(\theta') \tag{40}$$

Therefore $\boldsymbol{\alpha} = \boldsymbol{\alpha}'$. The same steps can be applied to proof the identifiability of $\boldsymbol{\beta}$ where the matrix diagonal $\boldsymbol{\alpha}$ is replaced by diagonal matrix of $\boldsymbol{\beta}$ where every $\beta_k, \forall k \in \{1, ..., K\}$ will be repeated $Q$ times before set $\beta_{k+1}$. It leads to a matrix with the same dimension $Q \times K$.

For $\boldsymbol{\Pi}$, let's define

$$U_{ij} = R(\boldsymbol{\theta})\boldsymbol{\beta}(\boldsymbol{\theta})\boldsymbol{\alpha}(\boldsymbol{\theta})\boldsymbol{\Pi}\boldsymbol{\alpha}(\boldsymbol{\theta})\boldsymbol{\beta}(\boldsymbol{\theta})(R(\boldsymbol{\theta}))^T$$

From previously, $R(\theta) = R(\theta')$, $\boldsymbol{\alpha}(\boldsymbol{\theta}) = \boldsymbol{\alpha}(\boldsymbol{\theta}')$ and $\boldsymbol{\beta}(\boldsymbol{\theta}) = \boldsymbol{\beta}(\boldsymbol{\theta}')$ then

$$U(\boldsymbol{\theta}) = U(\boldsymbol{\theta}') \rightarrow \boldsymbol{\Pi} = \boldsymbol{\Pi}' \tag{41}$$

$\square$

# B   Consistency of Maximum likelihood

*Proof.* The asymptotic consistency of the maximum likelihood estimator of Bernoulli uni layer SBM has been studied in Celisse et al. (2012); Bickel et al. (2013). The proof of the consistency of MDCSBM is straightforward from the proof of MSBM model, which can derived from the proof of uni-layer SBM. The proof can be performed by the same steps by taking

$$M_n(\pi, \alpha, \beta) = \frac{1}{N(N-1)L*K}$$

$$\sum_{l=1}^{L} log\Big(\sum_k \sum_{z_{[n]}} \beta_k \prod_{i \neq j} \mathbf{Bernoulli}(\pi_{z_i^k, z_j^k}^k) \prod_i \alpha_{z_i^k}^k\Big) \tag{42}$$

$$M(\pi) = \max_{a_{i,j} \in \mathcal{A}} \sum_k \sum_{q,w} \beta^{*k}\alpha_q^{*k}\alpha_w^{*k}$$

$$\sum_{q',w'} a_{qq'}^{*k}a_{ww'}^{*k}[\pi_{q,w}^{*k}log(\pi_{q',w'}^k) + (1 - \pi_{q,w}^{*k})log(1 - \pi_{q',w'}^k)] \tag{43}$$

where $\mathbf{Bernoulli}(\pi)$ is the Bernoulli distribution with parameter $\pi$, and $\beta^*, \alpha^* and \ \pi^*$ denote the true parameters respectively, with

$$\mathcal{A} = \{(a_{ij}^k)_{1 \leq q,w \leq Q^k}, a_{qw}^k \geq 0, \sum_w a_{qw}^k = 1\} \tag{44}$$

$\square$

