# OpenReview forum: "Mixture Degree-Corrected Stochastic Block Model for Multi-Group Community Detection in Multiplex Graphs"
_TMLR — Accepted by TMLR_

### Review · Reviewer_tnnY · 2024-09-25

**Summary Of Contributions:**

1. This paper proposes the "Mixture Degree-Corrected Stochastic Block Model" (MDCSBM), which aims to cluster similar layers by their community structures while simultaneously identifying communities within each layer's group.
2. This paper provides a rigorous definition of the model and uses an iterative technique for inference computations.
3. The effectiveness of the method is assessed using both real-world datasets and synthetic graphs.
4. This paper evaluates the identifiability of the proposed model and demonstrates the consistency of the maximum likelihood function through analytical analysis in supplementary materials

**Audience:**

Yes

**Claims And Evidence:**

Yes

**Requested Changes:**

1. Please review the format of the paper, paying special attention to the misuse of mathematical symbols, unclear mathematical representations, and the layout of figures and tables.
2. There are also unclear analysis in the experimental section. Explanations and descriptions of the figures and tables are unclear. Specifically, real-world data is mentioned again in Section 5.3 at the end of the experiments; how does this differ from Section 5.1? What are the experimental results, figures, and tables in this part?
3. Does the analysis of the algorithm's consistency in the appendix appear to be a simple extension of prior work? If not, please explain what technical innovations are present.
4. Please provide clear definitions or explanations of the metrics used in the experiments, such as BIC, and offer a more detailed introduction of the datasets used in the experimental section.

**Strengths And Weaknesses:**

Strengths:
The theoretical analysis of the model definition and variable estimation in this paper is logically clear, and the article includes both theoretical and experimental content, making it relatively comprehensive.


Weaknesses:
1. The main contribution of grouping layers and partitioning vertices is not novel, as it was previously analyzed in "Clustering Network Layers with the Strata Multilayer Stochastic Block Model" by Natalie Stanley et al. (IEEE Transactions on Network Science and Engineering, 2016).
2. While the paper introduces a new statistical model and employs algorithms with strong theoretical foundations, it lacks an analysis of the theoretical performance of these algorithms.
3. The main theoretical results are provided in the appendix, but they seem to be direct extensions of previous work without demonstrating technical difficulties or innovative points in the analysis.
4. As a theoretical model with strong assumptions, the experiments in this paper mainly focus on analyzing the proposed algorithm, with comparisons to other methods conducted only on simulated datasets, lacking substantial practicality analysis. And the analysis of experimental results is confusing.

Overall, the paper lacks sufficient theoretical depth, and the proposed algorithm does not possess significant practical value.

---

> ### Author Response · Authors · 2024-11-11
> **Response 1**
>
> We sincerely thank the reviewer for their insightful and thoughtful remarks. We greatly appreciate the time and effort they dedicated to meticulously reviewing our paper. The reviewer mentioned the work by Natalie Stanley et al., titled Clustering Network Layers with the Strata Multilayer Stochastic Block Model. After carefully reviewing their study, we acknowledge that while their approach is somewhat related to ours, our proposed work represents a significantly enhanced technique. Specifically, we relax the strong assumptions of the SBM model, which only accounts for uniform distributions within and between clusters, thus limiting its applicability in real-world scenarios. Our Degree-Corrected model allows for the optimization of clusters with non-uniform distributions, making it more flexible and applicable. Consequently, our model relies on less restrictive assumptions compared to previous approaches. Additionally, we contribute a theoretical analysis of our model, which was not present in earlier work, and propose an efficient initialization method that accelerates computation time and improves solution quality. In this revised version of the paper, we have included new experiments that demonstrate the benefits of using our proposed initialization model in terms of both performance and computational efficiency. We also reference Stanley’s model in the paper and provide a clear explanation of how our approach differs.
>
> Additionally, our theoretical results build on the foundational work cited in [1]. In our paper, we outline the main steps and principles necessary to understand the demonstration, providing a novel proof of consistency for multi-group community detection in multiplex graphs. We also demonstrate how to incorporate the group parameter ($\beta$) into our model. Furthermore, we propose a new method for identifiability that specifically accounts for the multiplex graph structure.
>
> Regarding the experimental analysis, it is important to note that the datasets we used do not contain ground truth labels or expert annotations. This limitation makes it challenging to fairly determine the "best" results when comparing models. Nonetheless, we have revised the experimental section to include more comprehensive details about each dataset and the key aspects to consider. We have also added comparisons between our model and the initialization method, an additional experiment to contrast the performance between random and proposed initializations, and references to prior studies that employed these real-world datasets.
>
> 1. We'd like to thank the reviewer for the comments, we're revising the mathematical notation of the whole document, we're also changing the size and layout of the figure as you recommend in this new version of the paper.
>
> 2. We apologize for the repeated mention of the experimental section and have corrected this in the new version of the paper. Regarding the results in the experimental section, we conducted an analysis of our model using two real-world datasets: brain scans and the Reality Mining study. We presented the intriguing results obtained and provided interpretations based on our findings. It is difficult to compare the performance and the results of our algorithm with other due to the absence of truth labels and expert analysis. Therefore, we did compare the results of our model with those from previous studies that utilized these datasets, particularly the brain scan data. Our model demonstrated superior performance, outperforming previous works, including [2], which employed a semi-supervised approach, and [3], which simplified the problem by aiming to find only two clusters—an approach that is inherently easier than our more complex model.
>
> 3. In the paper, we clarify the main concepts and lines that aid in understanding the consistency of our algorithm. Our work builds upon the results established in [1].  On the other hand, we would like to emphasize that we conducted a thorough analysis to establish the identifiability of our algorithm. This proof advances the state of the art by identifying new bounds of convergence for the consistency analysis and incorporating the parameter $\beta$ within the both proofs of consistency and indentifiability, which, to the best of our knowledge, is novel in this modeling framework.
>
> 4. The BIC (Bayesian Information Criterion) metric is used to determine the optimal number of clusters within a graph. In the revised paper, we have added a detailed explanation of the BIC formula and the method we used for its computation in our specific context in the supplementary materials. For performance comparison on synthetic datasets, we employed the NMI and AMI metrics. Since the tests algorithms provide a unified clustering result across all layers, we computed and reported the mean NMI/AMI scores for those proposed models to give an overall measure of performance.

---

> ### Author Response · Authors · 2024-11-11
> **Response 2**
>
> [1]:Alain Celisse, J. J. Daudin, and Laurent Pierre. Consistency of maximum-likelihood and variational estimators in the stochastic block model, 2012.
>
> [2]: Jesús Arroyo, Avanti Athreya, Joshua Cape, Guodong Chen, Carey E. Priebe, and Joshua T. Vogelstein. Inference for multiple heterogeneous networks with a common invariant subspace, 2020
>
> [3]: Anastasia Mantziou, Simon Lunagomez, and Robin Mitra. Bayesian model-based clustering for populations of network data, 2023.

---

> > ### Comment · Reviewer_tnnY · 2024-12-06
> >
> > Thank you for your response. Most of my concerns have been addressed in the revised version. However, I would still suggest that the authors refine the writing, particularly by clarifying the contributions of the paper from both theoretical and experimental perspectives. Currently, the description of the paper's main contributions is not very clear, and there are some redundant paragraphs (for instance, the fifth and last paragraphs of Section 1, as well as the last paragraph of Section 2, appear to convey similar ideas). Moreover, the contributions and insights regarding the experimental work are not introduced in the first two sections. We recommend that the authors consider adding a dedicated summary of the paper's contributions. Additionally, for the two theorems in the appendix, I suggest including them in the main text while keeping only their proofs in the appendix.

---

> > > ### Author Response · Authors · 2024-12-12
> > > **Response for the Official Comment by Reviewer tnny**
> > >
> > > Thank you very much for your feedback. We have updated and uploaded a new version of the document following all your comments. Best regards.

---

### Review · Reviewer_FFDE · 2024-10-23

**Summary Of Contributions:**

The authors consider the problem of community detection within multiplex graphs.
A multiplex graph is made of a set of nodes, and several layers, each layer being a set of edges between the nodes.
The considered task is to detect communities, ie define partitions of the nodes.

Previous approaches are divided by the authors into "unified communities", in which a single partition is found and shared across all layers, and "multilayer communities", in which a community can span several layers and a node belongs to a single community for a given layer, but can belong to a different community in another layer.

The authors propose a third kind of community detection, where layers are clustered into groups, then unified communities are defined for each group.
This approach is named "multi-group communities".
It thus requires a dual clustering to (i) cluster layers into groups and (ii) for a given cluster of layers, cluster the nodes into blocks (ie find a partition).
The authors formalize this model, describe an algorithm for identifying the groups and the blocks, and perform experiments on real and synthetic data.

**Audience:**

Yes

**Claims And Evidence:**

Yes

**Requested Changes:**

Overall the method and the experiments are well described so I have mostly minor comments.

In Eq.3 why does Θ not appear on the left hand-side?

Above Eq. 6 bold minuscule θk seems to be used to refer to the degree heterogeneity parameter of group k which is elsewhere denoted with a majuscule Θk?

In section 4.2, should "vertex-to-group" be "vertex-to-block" instead?

In 4.2.2: "writhed" → "written"

Section 4.3 states that the optimization algorithm for the initialization is described in the supplementary material but it is actually in section 4.4

In section 5.1.1 the proposed algorithm (almost) correctly groups multiple dMRI scans from each of 30 subjects (only collapsing 3 pairs of subjects and never placing scans from a single subject in different groups). While this is a good sanity check, being a structural imaging technique we expect dMRI to be somewhat stable for a given individual. It would be useful to also apply other methods for multiplex graph partitioning to this dataset to see if they succeed or fail to identify individuals. The following points are suggestions for future work rather than requests for this paper: for a more challenging task, it could be worth considering a dataset that has several populations (eg patients for a certain disease and controls) and seeing if the algorithm can group individuals who share some characteristic such as a diagnosis. Finally, the proposed method focuses not only on clustering layers into groups but also vertices into blocks. The experiment is done with a fixed number of 2 blocks (one per hemisphere). It could be interesting to increase the number of blocks and explore the identified partitions, to see if the algorithm identifies densely connected networks.

In section 5.2.8 it would be useful to compare the time complexity to that of other methods from the literature.

In section 5.2.9 a reference to an equation is broken (??)

I did not understand section 5.3 "real world data". It seems to be truncated and starts to describe experiments that are not mentioned elsewhere in the paper.

**Strengths And Weaknesses:**

Strengths:

- The method and the experiments are well described.
- The considered problem has a wide range of applications
- The proposed solution seems to identify meaningful layer groups and vertex blocks on real graphs, and its computation requirements seem low enough to make it practically useful.

Weaknesses:

- There are some typos and an incomplete section (see changes) so the paper needs another pass of proofreading
- The experiments on real data are somewhat limited. For the first experiment, identifying individuals from diffusion MRI scans is of limited usefulness and the approach is not compared to other methods. In the second method, the validation is only a qualitative assessment of the identified clusters and the approach is not compared to other methods.

---

> ### Author Response · Authors · 2024-11-11
> **Response 1**
>
> We thank the reviewer for their insightful comments and appreciate the understanding they have shown regarding our work, which motivates us to advance our research further. The reviewer raised important concerns about testing with real-world datasets, and we fully understand the nature of these questions. We would like to draw attention to some of the challenges we faced in this aspect of the study.
>
> Firstly, the datasets we used do not contain ground truth labels or expert analyses that provide detailed information about the class of each individual or their specific characteristics. Therefore, comparing the clustering results between different models without ground truth reference labels would not be entirely fair.
>
> Nonetheless, we provided a comparison of our algorithm with the model presented in [1], which is a semi-supervised approach, and with [2], which we outperformed, considering the simplifications made in their work. We hope this clarifies our approach and the rationale behind our methodological choices.
>
> Requested Changes:
>
> 1.  Thanks for your comment, we forgot to include it - a typo. We'll correct it in the new version.
>
> 2. The problem comes from the error of using $\boldsymbol{\theta}$ instead of $\boldsymbol{\Delta}$ in equation 5 and the text just after this equation, we correct the problem now by managing these issues.
>
> 3. Exactly, and we thank the appraiser for his foresight.
>
> 4. Thank you again.
>
> 5. Thank you for your attention. We fix the problem now.
>
> 6.  We thank the reviewer for their valuable remarks and acknowledge their expertise in the field of dMRI imaging and medical research. We agree with the observation that healthy individuals tend to exhibit stable brain connectivity patterns, leading to consistent clustering outcomes, which aligns with our findings. We also concur with the suggestion to apply our approach to datasets containing pathological cases in future work to evaluate the model's capacity to detect and differentiate disease-related patterns. We must also acknowledge our limited expertise in the medical field and are grateful for the reviewer's guidance in this regard. While we did not present a detailed comparison with other algorithms, this revised version of the paper includes a comparison with the studies in [1,2]. In the work by [1], our performance is comparable to the results achieved by their semi-supervised method, despite it involving extensive hyper-parameter tuning. Also, our algorithm outperforms in the model presented in [2], their results, even though they simplified the problem to clustering all layers into only two groups. Despite this simplification, their approach did not succeed in grouping images from each subject consistently within the same cluster. We hope this clarifies our contributions and highlights the novelty of our approach.
>
> 7. We acknowledge that presenting the time complexity of the other algorithms would be beneficial. In this revised version of the paper, we include a comparison of the time complexity of our model with the models used in the study. Our findings demonstrate that our model converges faster than all the others and scales more efficiently for larger networks.
>
> 8. we're correcting this problem and adding a small section on how to calculate the BIC value for this type of graph.
>
> 9. It's a little added text that we fix it in the this new version, thanks for the comment.

---

> > ### Author Response · Authors · 2024-11-11
> > **References**
> >
> > [1]: Jesús Arroyo, Avanti Athreya, Joshua Cape, Guodong Chen, Carey E. Priebe, and Joshua T. Vogelstein. Inference for multiple heterogeneous networks with a common invariant subspace, 2020
> >
> > [2]: Anastasia Mantziou, Simon Lunagomez, and Robin Mitra. Bayesian model-based clustering for populations of network data, 2023.

---

> > > ### Comment · Reviewer_FFDE · 2024-11-12
> > >
> > > Thank you for answering my questions and updating the manuscript. The issues I found in the first version have all been addressed in this revision

---

### Review · Reviewer_sYid · 2024-10-29

**Summary Of Contributions:**

The authors propose an extension of the Degree-Corrected Stochastic Block Model (DCSBM) for multiplex graphs. The graphs in different layers are partitioned into K groups, and the vertices of a given group are partitioned into a group-specific number of communities. The model is essentially a mixture distribution of K independent DCSBMs. This kinds of models are known to be sensitive to initialisation, thus, the authors propose an alternating optimisation procedure to find a good initialisation.

**Audience:**

Yes

**Broader Impact Concerns:**

No concerns.

**Claims And Evidence:**

Yes

**Requested Changes:**

- By "Estimation Maximization (EM)" and Variational EM do you mean "Expectation Maximization"? If yes, please use the established terminology.
- What is the performance after initialisation (Alg. 2) before running Alg. 1? In other words, the initialisation strategy can serve as a very good baseline for comparison. This can be added to all synthetic experiments.
- Please provide more details on the experimental evaluation. How is NMI/AMI computed? Is it the average of the NMI/AMI on groups + NMI/AMI on communities?
- Please provide more details on the baselines. Since based on the exposition none of the baselines seem to be directly designed for multi-group communities how are the used in this setting?
- Can you provide the code to reproduce the experiments?
- A lower priority but nonetheless interesting addition would be to compare the proposed initialisation to random initialisation.

**Strengths And Weaknesses:**

Given the assumptions in the paper the extension is natural and well-formulated. The approach addresses a different niche -- multi-group communities -- compared to existing approaches for multiplex graph. The presentation is clear and the initialisation strategy is reasonable. The experimental evaluation on synthetic data confirms that when the data is indeed generated from the proposed model the algorithm can infer the ground-truth groups and communities. In contrast, the evaluation on realistic data is less convincing. For example the brain connectivity dataset seems relatively easy and is likely closer to a "toy" dataset. The data in the reality mining study seems more suited for temporal approaches (the authors assume each layer is one day). Overall, it is not very convincing that there is a "need" for such a model in the real world, or that real-world graphs are likely to follow a mixture of DCSBMs.

---

> ### Author Response · Authors · 2024-11-11
> **Response 1**
>
> We would like to sincerely thank the reviewer for their insightful comments and feedback. We appreciate the reviewer's concern regarding the limitations associated with real-world data experiments and would like to clarify any potential ambiguity related to this issue. The datasets used in our study are open-source and do not include ground truth labels validated by experts. As a result, given that our analysis relies on unsupervised learning methods, it is challenging to evaluate which model performs better without expert validation and labeled data.
>
> Regarding brain connectivity data, we recognize that the dataset may appear simpler than it is. The complexity primarily arises from the structural and topological properties of the data. Brain signals can display structural similarities between patients, potentially leading to interference effects in results. However, we did not encounter such interferences in our analysis. We conducted a comparison of our results with previous studies and found that our method performed comparably to the semi-supervised model presented in [1] and outperformed the unsupervised model in [2]. We add this analysis in the new version of the paper. Furthermore, this experiment is valuable as it could, in some instances, facilitate the identification of shifts in patient health status by detecting transitions from one cluster to another. However, it is important to note that we cannot validate this hypothesis within our current study, as all participants in the dataset are reported to be healthy.
>
> Regarding the Reality Mining study, although each layer represents a single day, it is feasible to model this data as a multiplex graph. A temporal graph can indeed be seen as a type of multiplex graph where the layers are organized sequentially. To the best of our knowledge, there are no existing temporal models that directly address the specific objectives of our research.
>
> Moreover, the proposed multi-group community detection approach is applicable across various fields. For instance, in the domain of electrical grids, which consist of physical infrastructures for transmitting electrical energy from producers to consumers, effective management is increasingly complex an incertitude due to the high penetration of renewable energy sources. A viable solution involves segmenting the grid into smaller zones, each managed independently. Since the state of the grid continuously changes based on fluctuations in energy consumption and generation and the changes in the topology of the grid, these zones must also adapt dynamically. Multi-group community detection can be advantageous here, as it allows for adaptive segmentation based on varying grid conditions, such as high wind energy penetration or fluctuations in solar energy availability, treating each state distinctly yet consistently across similar scenarios.
>
> This example illustrates a practical application of our proposed algorithm. We would be happy to elaborate further or provide additional examples if necessary.
>
> Requested Changes:
>
> 1. Indeed, we meant Expectation Maximization, thank you, and we're correcting that in the new version.
> 2. We would like to thank the reviewer for their insightful remark. In response, we have added a new experiment in which we compare the performance of the initialization model based on spectral representation and the proposed MDCSBM model using a synthetic dataset. Our findings indicate that running the MDCSBM results in improved performance, highlighting that the MDCSBM acts as an additional refinement step that further enhances the initial results provided by the spectral representation. Moreover, the primary objective of the paper was to develop a generative model capable of simultaneously determining clusters and generating data that adhere to the same distribution as the original dataset.
>
> 3.Once the optimization of communities is complete on each model, we calculate the NMI/AMI for each layer independently. By evaluating the error on each individual layer and then averaging these values, we provide a mean performance metric across all layers. This approach enables us to present a comprehensive measure of performance across the entire dataset, effectively illustrating the model's efficacy over all layers. We add this explanation on the new version of the paper.
>
>
> [1]: Jesús Arroyo, Avanti Athreya, Joshua Cape, Guodong Chen, Carey E. Priebe, and Joshua T. Vogelstein. Inference for multiple heterogeneous networks with a common invariant subspace, 2020
> [2]: Anastasia Mantziou, Simon Lunagomez, and Robin Mitra. Bayesian model-based clustering for populations of network data, 2023.

---

> > ### Author Response · Authors · 2024-11-11
> > **Response 2**
> >
> > 4. The baseline methods we used are designed for multiplex graph community detection, just like our proposed model. However, as noted, these baselines are not intended for applications involving multi-group community detection. Once these baseline models optimize the communities, they are constrained to use the same set of communities across all layers. This constraint highlights a potential limitation: if the community structures differ across layers, the optimized communities could be suboptimal for representing the overall structure of the data. Our experiments underscore the importance of verifying whether all layers share the same community structures. If they do not, the optimized communities may indeed fail to capture the complexity of the data accurately. One might ask why we did not use algorithms specifically designed for multi-group community detection. Unfortunately, to the best of our knowledge, at the time of writing this paper, no such algorithms were founded for a direct comparison. In summary, our comparison with these baseline methods is not solely to demonstrate the superior performance of our model. Rather, it also serves to emphasize that, in scenarios where the community structures vary between layers, using conventional multiplex community detection methods could yield suboptimal results. We explain the setting of the used algorithm within the paper.
> >
> > 5. The codes are available but not publishable in the current state, they need to be refined, commented... We will publish the codes and attach them to the article once it has been published. If the reviewer absolutely needs the code, we can send it via the journal member community.
> >
> > 6. This is indeed a crucial question, as the performance of the proposed algorithm is highly dependent on initialization. To address this, we have included a new experiment comparing the random initialization with our proposed initialization model. We observed that the proposed initialization model not only converges faster but also delivers better performance compared to the random initialization, which requires more time and yields suboptimal results. This underscores the importance and effectiveness of the proposed initialization strategy.

---

> > > ### Comment · Reviewer_sYid · 2024-11-14
> > > **Thank you**
> > >
> > > Thank you for including the additional experiments and the additional explanation.

---

### Author Response · Authors · 2024-11-11
**New Version of Article**

Dear all,

We have revised the paper to incorporate all the comments and suggestions provided by the reviewers. We kindly invite the reviewers to review this updated version.

Thank you for the time and effort you've dedicated to evaluating our work.

---

### Decision · Action_Editor_LtD3 · 2025-01-11

**Recommendation:** Accept as is

**Comment:**

All of reviewers feel that the paper provides a contribution to the problem of community detection. The problem is well formulated and the approach addresses a different niche (multi-group communities). Some of concerns raised by reviewers were well resolved. Although there are still some concerns about experiments on real-world data, I believe the paper is deserved to be published.

**Audience:**

The problem considered in this paper has wide range of applications. Multi-group communities and the extension addressed in this paper could be interesting for people working on stochastic block models.

**Claims And Evidence:**

This paper addresses the problem of community detection within multiplex graphs. The approach taken in this paper is "multi-group communities", which requires dual clustering to cluster layers into groups as well as to cluster the nodes into blocks for a given clusters of layers. The method and the experiments are well described. The experiments on both synthetic and real-world data are provided to justify the method. Although the experiments on real-world data is somewhat limited, a single paper cannot include everything.